# SARS-CoV-2 Omicron variant causes mild pathology in the upper and lower respiratory tract of hamsters

Federico Armando [1,6], Georg Beythien [1,6], Franziska K. Kaiser [2,6], Lisa Allnoch [1], Laura Heydemann [1], Malgorzata Rosiak [1], Svenja Becker[1], Mariana Gonzalez-Hernandez [2], Mart M. Lamers [3], Bart L. Haagmans [3], Kate Guilfoyle[4], Geert van Amerongen[4], Malgorzata Ciurkiewicz [1,6], Albert D.M.E. Osterhaus[2,5,6] & Wolfgang Baumgärtner [1,6 ✉]

Since its discovery in 2019, multiple variants of severe acute respiratory syndrome coronavirus-2 (SARS-CoV-2) have been identified. This study investigates virus spread and associated pathology in the upper and lower respiratory tracts of Syrian golden hamsters at 4 days post intranasal SARS-CoV-2 Omicron infection, in comparison to infection with variants of concern (VOCs) Gamma and Delta as well as ancestral strain 614 G. Pathological changes in the upper and lower respiratory tract of VOC Omicron infected hamsters are milder than those caused by other investigated strains. VOC Omicron infection causes a mild rhinitis with little involvement of the olfactory epithelium and minimal lesions in the lung, with frequent sparing of the alveolar compartment. Similarly, viral antigen, RNA and infectious virus titers are lower in respiratory tissues of VOC Omicron infected hamsters. These findings demonstrate that the variant has a decreased pathogenicity for the upper and lower respiratory tract of hamsters.

[1] Department of Pathology, University of Veterinary Medicine, Foundation, Hanover, Germany. [2] Research Center for Emerging Infections and Zoonoses, University of Veterinary Medicine, Foundation, Hanover, Germany. [3] Department of Viroscience, Erasmus MC, Rotterdam, Netherlands. [4] Viroclinics Xplore, Schaijk, Netherlands. [5] Global Virus Network, Center of Excellence, University of Veterinary Medicine, Foundation, Hanover, Germany. [6]These authors contributed equally: Federico Armando, Georg Beythien, Franziska K. Kaiser, Malgorzata Ciurkiewicz, Albert D.M.E. Osterhaus, Wolfgang Baumgärtner. ✉email: Wolfgang.Baumgaertner@tiho-hannover.de

Since the first case of COVID-19 was reported more than 330 million SARS-CoV-2 infections and 5.5 million deaths have been detected worldwide. After the initial identification of the SARS-CoV-2 strain coming from Wuhan, several variants of concern (VOCs) have been identified including Alpha (B.1.1.7) in the United Kingdom, Beta (B.1.351) in South Africa, Gamma (P.1) in Brazil, and Delta (B.1.617.2) in India. These were associated with decreased effectiveness of medical interventions, epidemiological changes and/or a variable clinical course, collectively necessitating adjustment of public health and societal measures[1,2]. Recently, a new SARS-CoV-2 variant was identified in Botswana and later in South Africa, that was designated by the World Health Organization (WHO) VOC Omicron (B.1.1.529), based on an observed, unusually high number of mutations, high transmissibility and partial escape from pre-existing immunity[3–5]. First reports about clinical and epidemiological features of the infection indicate a relatively mild disease course and increased human-to-human virus spread[6]. So far, the potentially reduced pathogenicity of the VOC Omicron has not been addressed thoroughly in vivo experimental studies[7]. The WHO has officially declared that the first clinical data of VOC Omicron cases indicate its association with relatively mild disease[8], but these data have not yet been corroborated by in-depth pathology studies.

Moreover, information about the relative pathogenicity and epidemiological significance of VOC Omicron compared to previous VOCs, which should be a major guide to public health response, is also urgently needed. Besides the implementation of non-pharmaceutical intervention protocols, this includes the development of future vaccination and antiviral treatment strategies for the ongoing pandemic. Among the experimental animal models, the use of non-human primates or ferrets, generally offering effective models for pre-clinical evaluation of vaccines and therapeutics for respiratory infections, provide limited options for pathogenicity studies. These models mimic asymptomatic to mild COVID-19 clinical courses only and therefore should not be considered the first choice to assess VOC Omicron pathogenicity and test COVID-19 vaccines and therapeutics[9–13]. The Syrian golden hamster (Mesocricetus auratus) model on the other hand, has been shown to mimic moderate to severe COVID-19[14] and has been proven useful for evaluation of COVID-19 vaccines and therapeutics[15], and recently for assessment of the pathogenicity of SARS-CoV-2 variants, such as VOC Omicron[16].

The present study aims to obtain crucial information on the pathogenicity of VOC Omicron, focusing specifically on the respiratory tract. For this reason, the upper and the lower respiratory tract of 8–10 weeks old Syrian golden hamsters were thoroughly investigated following intranasal inoculation with $10^4$ TCID$_{50}$ of either SARS-CoV-2 614 G, or VOC Gamma (P.1, Brazil), VOC Delta (B.1.617.2, India) or VOC Omicron (B.1.1.529, South Africa) variants. We show that, in contrast to other SARS-CoV-2 strains, VOC Omicron causes a subclinical infection in hamsters, which is characterized by a low viral load, mild pathology and decreased inflammatory cell infiltrates in the nasal cavity and lungs at 4 days post infection. The result confirm the lower pathogenicity of VOC Omicron compared to other SARS-CoV-2 strains in the hamster model and complement previous studies by providing a detailed analysis of histopathological lesions throughout the respiratory tract.

## Results

### VOC Omicron infection in hamsters does not cause weight loss and induces minimal macroscopic lesions in lungs.
Upon daily recording, hamsters infected with SARS-CoV-2 614 G or VOCs Delta and Gamma showed a body weight loss that reached an

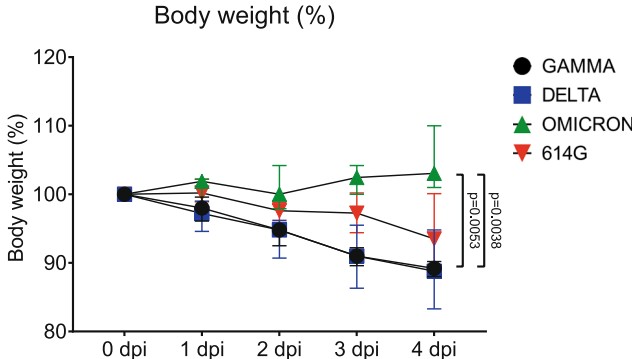

**Fig. 1 VOC Omicron infection in hamsters does not cause weight loss.** Severe acute respiratory syndrome coronavirus-2 (SARS-CoV-2) variants of concern, Gamma and Delta infected hamsters showed marked weight loss of 10.8% and 11.2%, respectively. SARS-CoV-2 614 G infected hamsters revealed a moderate weight loss of 6.5%, whereas Omicron infected hamsters showed a mild increase in body weight averaging 3.1%. Data points indicate the group median, error bars indicate minima and maxima. The difference was significant comparing VOC Omicron infected hamsters with VOC Gamma and with VOC Delta. Data was tested by two-tailed Mann–Whitney-U tests followed by Benjamini–Hochberg correction. A p-value of ≤0.05 was chosen as the cut-off for statistical significance. N = 5 animals/group for VOCs Gamma and Delta and 6 animals/group for VOC Omicron and 614 G. Source data are provided as a Source Data file. Dpi = days post infection.

average of 6.5%; 11.2% and 10.8% after 4 days, respectively. In contrast, SARS-CoV-2 Omicron infected hamsters showed an average increase in body weight of 3,1% (Fig. 1). Statistical analysis showed a significant difference in body weight between VOC Omicron infected and VOCs Delta and Gamma infected hamsters at 4 days post infection (dpi) (Supplementary Table 1).

At necropsy, 4 dpi with SARS-CoV-2 614 G and VOCs Gamma and Delta respectively, macroscopically the lungs of all hamsters showed moderate to severe, multifocal to coalescing areas of dark-red discoloration and consolidation with variable areas of pale discoloration on the surfaces. In contrast, the lungs of hamsters infected with the VOC Omicron showed only minimal foci of reddish discoloration. No other organs displayed macroscopic changes.

These results indicate that acute VOC Omicron infection of hamsters leads to milder macroscopic lung lesions compared to infection with SARS-CoV-2 614 G and VOCs Gamma and Delta, respectively.

### VOC Omicron infection in hamsters displays decreased pathogenicity for the upper respiratory tract.
In all SARS-CoV-2 infected hamsters, nasal turbinate lesions were observed. Microscopically, lesions were characterized by epithelial necrosis, neutrophilic/heterophilic exocytosis and sub-epithelial infiltration of macrophages and neutrophils/heterophils. The inflammation affected the respiratory and olfactory mucosa. Epithelial changes were more severe in the olfactory mucosa, which showed multifocal to coalescing areas of disorganization, necrosis and sloughing, associated with intraluminal exudate composed of proteinaceous fluid, cell debris and degenerated neutrophils/heterophils. In the respiratory mucosa, only occasional cell death, small amounts of exudate and a reactive hyperplasia were observed. Rhinitis was moderate to marked in the majority of hamsters infected with either SARS-CoV-2 614 G or VOCs Delta and Gamma. In contrast, hamsters infected with the VOC Omicron showed milder rhinitis. To confirm this, quantification of histopathological lesions was performed with a semi-

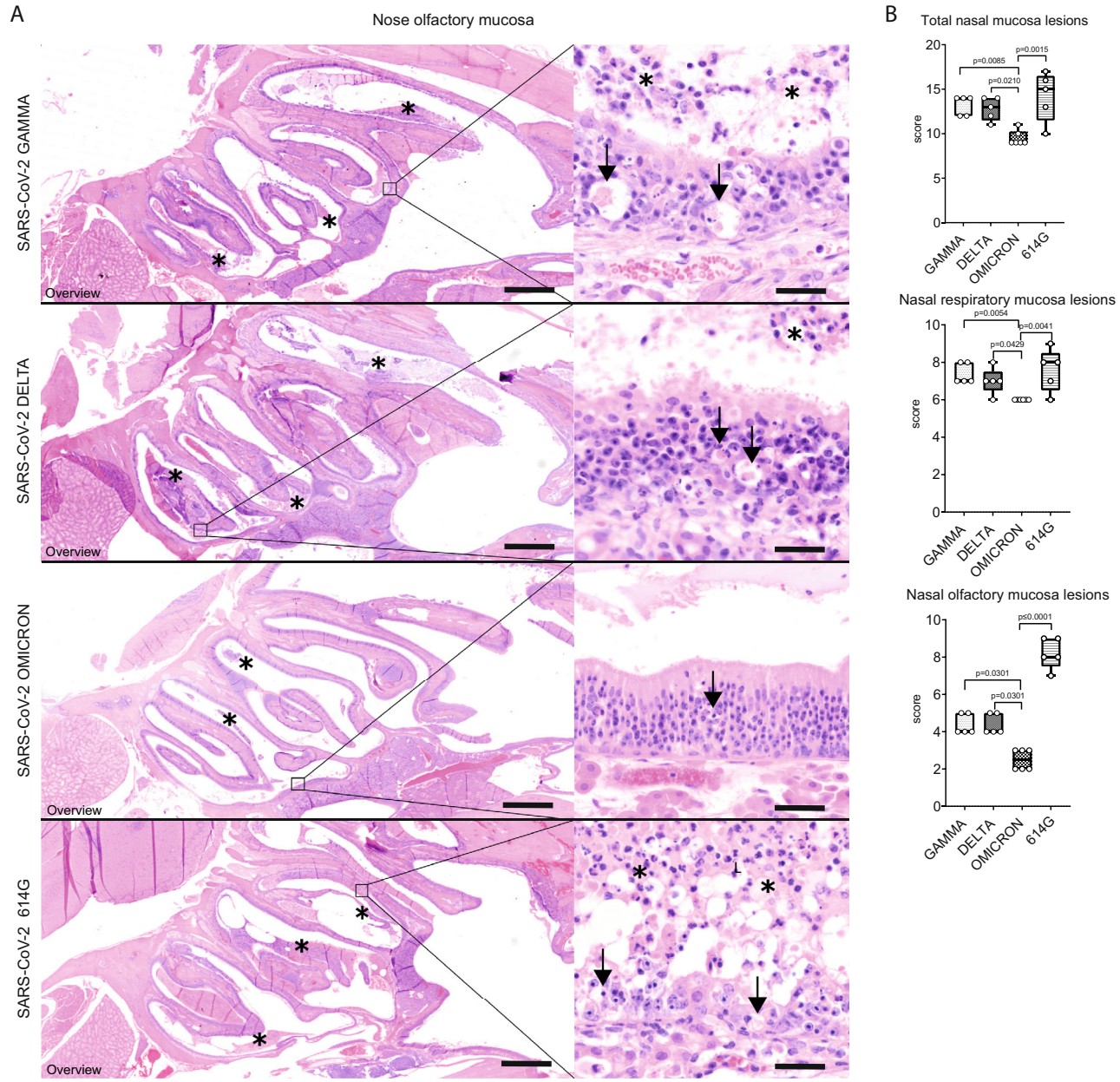

**Fig. 2 Decreased pathogenicity of VOC Omicron for nasal olfactory mucosa compared to other SARS-CoV-2 strains. A** Representative images showing an overview of the caudal nasal turbinates (left panel) and high magnification of the olfactory mucosa (right panel) of hamsters infected with VOCs Gamma, Delta, Omicron or 614 G strain. Olfactory mucosa in VOCs Gamma and Delta or 614G infected hamsters showed the most striking histopathological changes. Lesions were mainly characterized by disorganization, cell death (arrows) and cell sloughing, associated with intraluminal exudate composed of proteinaceous fluid, cell debris and degenerated neutrophils/heterophils (asterisks). Olfactory mucosa in VOC Omicron infected hamsters was largely intact and only occasional cell death was observed (arrow). Scale bars = 1 mm (overviews) and 50 µm (high magnifications). Hematoxylin and eosin stain. **B** Semi-quantitative analysis of nasal turbinates histopathology revealed significantly lower values in the overall score as well as the separate scores for respiratory and olfactory mucosa in VOC Omicron infected hamsters. Data are shown as box and whisker plots. The bounds of the box plot indicate the 25th and 75th percentiles, the bar indicates medians, and the whiskers indicate minima and maxima. Data were tested by two-tailed Mann–Whitney-*U* tests followed by Benjamini–Hochberg correction. A *p*-value of ≤0.05 was chosen as the cut-off for statistical significance. *N* = 5 animals/group for VOCs Gamma and Delta and 6 animals/group for VOC Omicron and 614 G. For quantification, one entire longitudinal section of the nasal turbinates was evaluated per animal. Source data are provided as a Source Data file.

quantitative scoring system, which includes the assessment of inflammation, hyperplasia, necrosis, and intraluminal cell debris. The overall nasal turbinate histopathological score of hamsters infected with VOC Omicron was significantly lower than that of hamsters infected with any other strain (Supplementary Table 2, Fig. 2). Separate pathological evaluation of the respiratory and olfactory mucosa revealed that scores were significantly lower in both anatomical compartments in VOC Omicron infected hamsters compared to all other groups (Supplementary Table 2, Fig. 2). Histopathological semi-quantitative scores of the nasal turbinates were mostly confirmed by the quantification of cells expressing myeloperoxidase (MPO, marker for neutrophils/heterophils) or ionized calcium-binding adapter molecule 1 (IBA1, marker for macrophages and dendritic cells). In particular, VOCs

Gamma and Delta infected hamsters showed significantly higher numbers of MPO+ cells in the total nasal mucosa compared to VOC Omicron infected hamsters. In the nasal respiratory mucosa, VOC Gamma infected hamsters revealed a significantly higher number of MPO+ cells compared to VOC Omicron infected hamsters. The nasal olfactory mucosa of VOC Omicron infected hamsters showed a lower number of MPO+ cells compared to VOCs Gamma and Delta as well as SARS-CoV-2 614 G infected hamsters. Interestingly, the respiratory mucosa of VOC Omicron infected hamsters showed a significantly higher number of IBA1+ cells, whereas the olfactory mucosa of VOC Omicron infected hamsters showed a significantly lower number of IBA1+ cells compared to all other groups (Supplementary Table 2, Supplementary Fig. 1).

In summary, VOC Omicron infected hamsters showed a significantly lower severity of lesions in the upper respiratory tract compared to those of the other investigated groups. This finding led to the expectation that also viral antigen expression and infectious virus titers in VOC Omicron infected animals would be lower.

Viral antigen was mainly detected in the olfactory epithelium of the respective groups, while only scattered epithelial cells were immunolabeled in the respiratory mucosa. Quantification of the viral antigen in the respiratory epithelium of the nasal turbinates revealed no significant differences among the groups. However, quantification of viral antigen in the olfactory epithelium of VOC Omicron infected hamsters proved to be the lowest among the investigated groups, with statistically significant difference when compared to SARS-CoV-2 614 G (Supplementary Table 2, Fig. 3). In addition, SARS-CoV-2 614 G infected hamsters showed significantly higher numbers of SARS-CoV-2 NP+ cells in the olfactory mucosa compared to VOC Delta (Supplementary Table 2).

Similar to the findings obtained by immunolabeling, infectious SARS-CoV-2 titers in the nasal turbinates were the lowest in VOC Omicron infected hamsters with statistically significant differences when compared to 614 G and VOC Gamma (Supplementary Table 2, Fig. 3). Interestingly, VOC Gamma infected hamsters showed significantly higher SARS-CoV-2 titers in the nasal turbinates than those infected with VOC Delta (Supplementary Table 2).

Quantification of SARS-CoV-2 RNA in the nasal turbinates confirmed the virus titration results. VOC Omicron infected hamsters showed the highest Ct values, corresponding to the lowest viral RNA copies. The Ct values were significantly higher compared to VOC Delta and Gamma. In addition, VOC Gamma infected hamsters showed significantly lower Ct values compared to 614 G infected hamsters (Supplementary Table 2, Fig. 3). Taken together, VOC Omicron infection displayed decreased pathogenicity for the upper respiratory tract of hamsters and was characterized by a mild rhinitis with reduced amount of viral antigen in the olfactory epithelium as well as decreased infectious SARS-CoV-2 titers and viral RNA load. This lower pathogenicity of VOC omicron for the upper respiratory tract prompted us to study its pathogenicity for the lower respiratory tract.

**VOC Omicron infection in hamsters induces a moderate inflammation with intralesional viral antigen expression in the trachea.** Tracheal lesions were observed in all SARS-CoV-2 infected hamsters and were characterized by multifocal to coalescing sub-epithelial infiltration with macrophages, lymphocytes and neutrophils/heterophils with frequent neutrophilic/heterophilic exocytosis. In addition, scattered single cell death and ciliary loss were observed. The severity of tracheitis varied from mild to moderate in individual animals, regardless of the SARS-

CoV-2 variant used. Quantification of histopathological lesions was performed with a semi-quantitative scoring system, which includes an assessment of inflammation, hyperplasia, epithelial degeneration/necrosis, and intraluminal cell debris. The quantification of tracheal histopathology by semi-quantitative scoring showed no statistically significant differences among the groups (Supplementary Table 3, Fig. 4). The results of histopathological scoring were confirmed by the quantification of immunolabeling for MPO+ and IBA1+ cells. For both cell markers, no statistically significant differences were observed among the groups (Supplementary Table 3, Supplementary Fig. 2). Based on these findings we further analyzed viral antigen expression in VOC Omicron infected animals comparing it with that of VOCs Gamma and Delta and SARS-CoV-2 614 G. Viral antigen was exclusively detected in VOC Omicron infected hamsters, resulting in a statistically significant difference compared to all other groups. However, the number of positive cells was below 5% of all epithelial cells in all animals of this group. (Supplementary Table 3, Fig. 5).

Taken together, the trachea of VOC Omicron infected hamsters showed a mild to moderate tracheitis similar to the other groups, despite slightly higher numbers of SARS-CoV-2 NP+ cells.

**VOC Omicron infection of hamsters displays a lower pathogenicity for the lower respiratory tract than previous VOCs.** All SARS-CoV-2 infected hamsters showed inflammatory lesions in the lung, but the distribution and severity varied among the groups. Histopathological lesions were characterized by multifocal to coalescing broncho-interstitial pneumonia. Alveoli were often obscured by septal and luminal infiltrates of macrophages and neutrophils/ heterophils, admixed with extravasated erythrocytes and fibrin. Conductive airways showed occasional cell death and mucosal infiltration with neutrophils/heterophils and macrophages. Conductive airway epithelia frequently also showed piling up of cells and increased number of mitoses, interpreted as hyperplasia. In addition to alveoli and conductive airways, histopathological lesions were also observed in the vascular compartment. These were characterized by histiocytic and neutrophilic/heterophilic vasculitis with vascular wall degeneration, endothelialitis and luminal endothelial cell proliferation, interpreted as endothelial hypertrophy and hyperplasia. In addition, perivascular lympho-histiocytic cuffs, perivascular edema, and perivascular hemorrhages were also frequent.

Pneumonia was moderate to marked in the majority of hamsters infected with either SARS-CoV-2 614 G or VOCs Delta and Gamma. Quantification of histopathological lesions was performed with a semi-quantitative scoring system, which includes an assessment of inflammation, hyperplasia, necrosis, edema, hemorrhage and fibrin exudation. Interestingly, a milder pneumonia was consistently observed in hamsters infected with VOC Omicron. The overall lung histopathological score of VOC Omicron infected hamsters was significantly lower compared to VOCs Gamma and Delta and SARS-CoV-2 614 G infected hamsters (Supplementary Table 4, Fig. 6). Separate evaluation of the lesions in alveoli, conductive airways and vessels showed that the scores were significantly lower in all three compartments of VOC Omicron infected animals compared to the other groups (Supplementary Table 4, Fig. 6). Interestingly, lesions in VOC Omicron infected hamsters seemed to be mainly centered on the airways and the vascular compartment, while alveoli showed no or minimal involvement. In contrast, hamsters infected with either SARS-CoV-2 614 G or VOCs Gamma and Delta showed equal involvement of all compartments. Quantification of MPO+ and IBA1+ cells largely confirmed histopathological scoring results. In particular, SARS-CoV-2 614 G and VOC Omicron infected hamsters showed a significantly lower number of MPO+

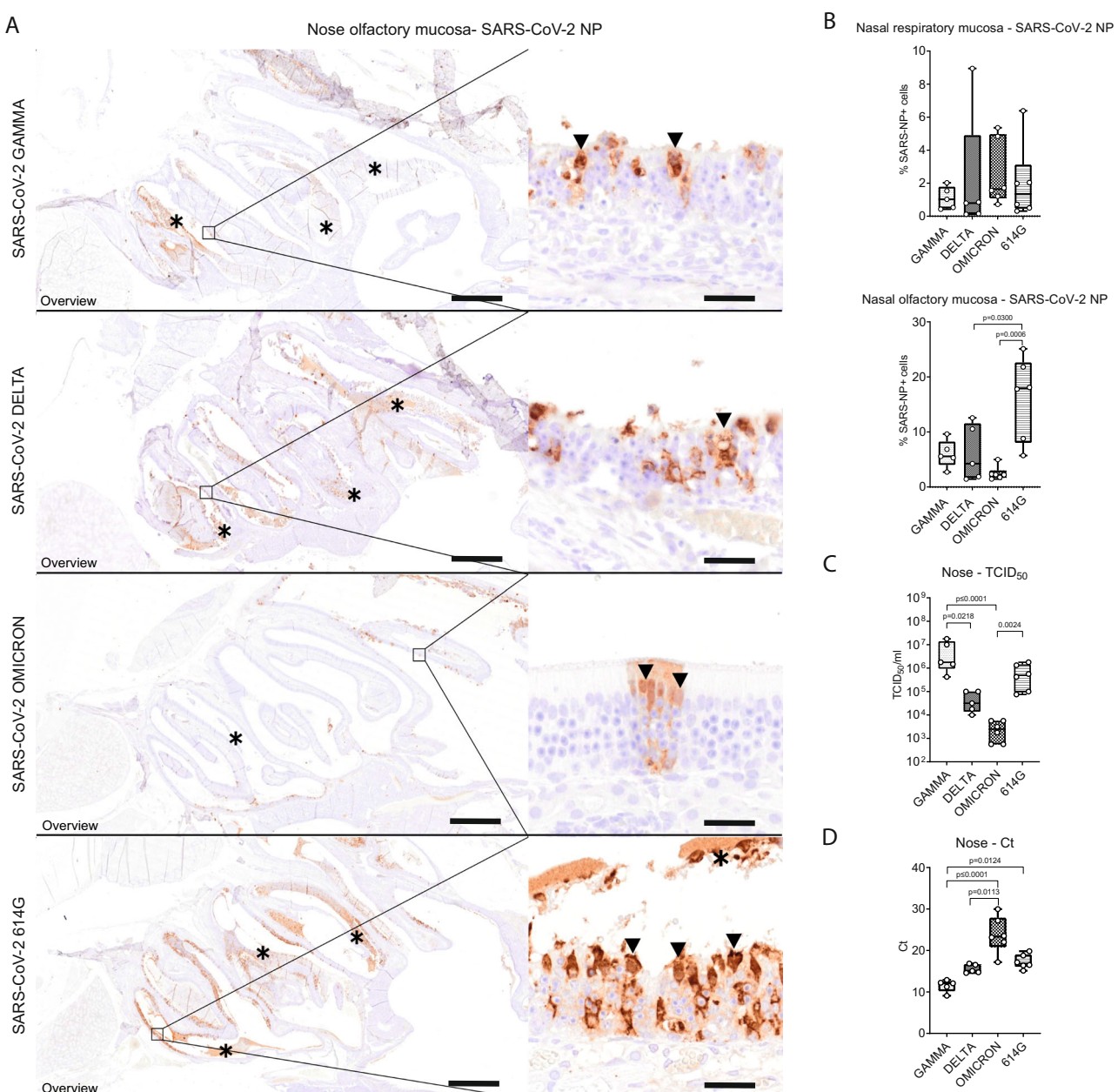

**Fig. 3 VOC Omicron infected hamsters showed decreased viral antigen, RNA and infectious viral titers in the nasal turbinates compared to other SARS-CoV-2 strains. A** Representative images showing SARS-CoV-2 nucleoprotein (NP) immunolabeling in the nasal turbinates of hamsters infected with VOCs Gamma, Delta, Omicron or 614 G strain. The images depict an overview of the caudal nasal turbinates (left panel) and high magnification of the olfactory mucosa (right panel). Viral antigen (brown signal) was mainly detected in epithelial cells within the olfactory mucosa and within intraluminal exudates (asterisks). Based on the cell morphology, characterized by apically located nuclei and abundant apical cytoplasm, most of these cells were sustentacular cells (arrowheads). The number of positive cells was highest in hamsters infected with 614 G, followed by hamsters infected with VOCs Gamma or Delta. In hamsters infected with VOC Omicron, only rare foci with a few positive cells were observed. Scale bars: 1 mm (overview) and 20 μm (high magnification). **B** Quantification of SARS-CoV-2 NP antigen in the respiratory and olfactory mucosa. VOC Omicron infected hamsters displayed a significantly lower number of immunolabeled cells in the olfactory mucosa compared to hamsters infected with 614G. **C** Quantification of infectious SARS-CoV-2 titers in the nasal turbinates. Titers were significantly lower in VOC Omicron infected hamsters compared to 614 G and VOC Gamma infected hamsters. **D** Quantification of SARS-CoV-2 RNA by qRT-PCR (Ct values). VOC Omicron infected hamsters show significantly higher Ct values, corresponding to low RNA content, compared to VOCs Gamma and Delta infected hamsters. **B–D** Data are shown as box and whisker plots. The bounds of the box plot indicate the 25th and 75th percentiles, the bar indicates medians, and the whiskers indicate minima and maxima. Data were tested by two-tailed Mann–Whitney-U tests followed by Benjamini–Hochberg correction. A *p*-value of ≤0.05 was chosen as the cut-off for statistical significance. $N = 5$ animals/group for VOCs Gamma and Delta and 6 animals/group for VOC Omicron and 614 G. For quantifications, one entire longitudinal section of the nasal turbinates was evaluated per animal. Source data are provided as a Source Data file. Ct = cycle threshold; $TCID_{50}$ = tissue culture infection dose 50.

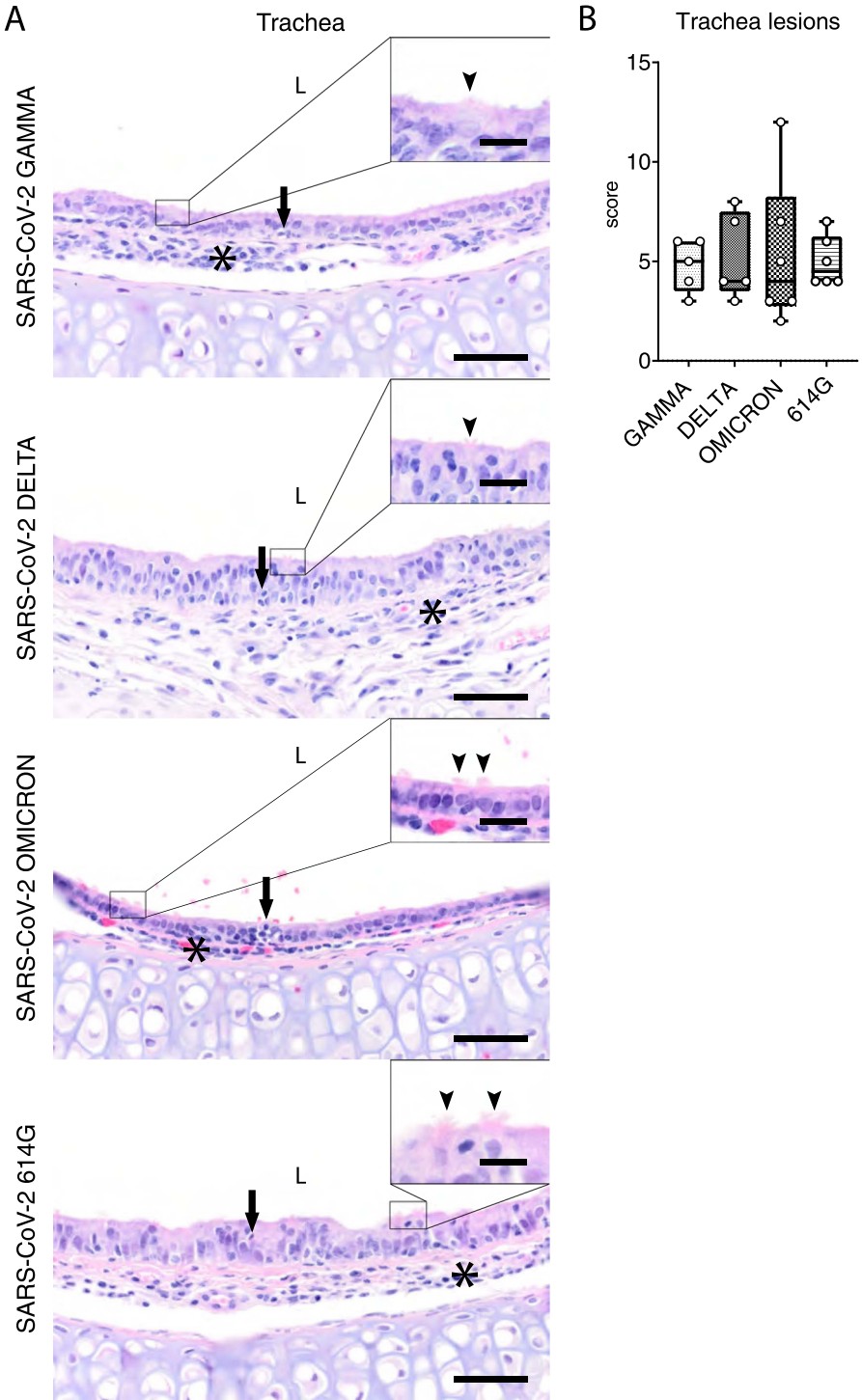

**Fig. 4 Tracheal lesions are mild to moderate regardless of the SARS-CoV-2 strain infection in hamsters. A** Representative images showing tracheal lesions in hamsters infected with VOCs Gamma, Delta, Omicron or 614 G strain. Trachea histopathological changes were characterized by neutrophilic/heterophilic exocytosis (arrows) and sub-epithelial infiltration of macrophages and neutrophils/heterophils (asterisks). The insets show a higher magnification of the ciliated epithelium. Intact cilia are indicated by arrowheads. In all groups, most cells show ciliary loss. Hematoxylin and eosin stain. Scale bars = 50 µm and 10 µm (inset). **B** Semi-quantitative analysis of tracheal histopathology revealed no statistically significant differences in the recorded scores among the groups. Data are shown as box and whisker plots. The bounds of the box plot indicate the 25th and 75th percentiles, the bar indicates medians, and the whiskers indicate minima and maxima. Data were tested by two-tailed Mann–Whitney-U tests followed by Benjamini–Hochberg correction. A p-value of ≤0.05 was chosen as the cut-off for statistical significance. N = 5 animals/group for VOCs Gamma and Delta and 6 animals/group for VOC Omicron and 614 G. For quantifications, one entire cross- and one entire longitudinal section were evaluated per animal. Source data are provided as a Source Data file.

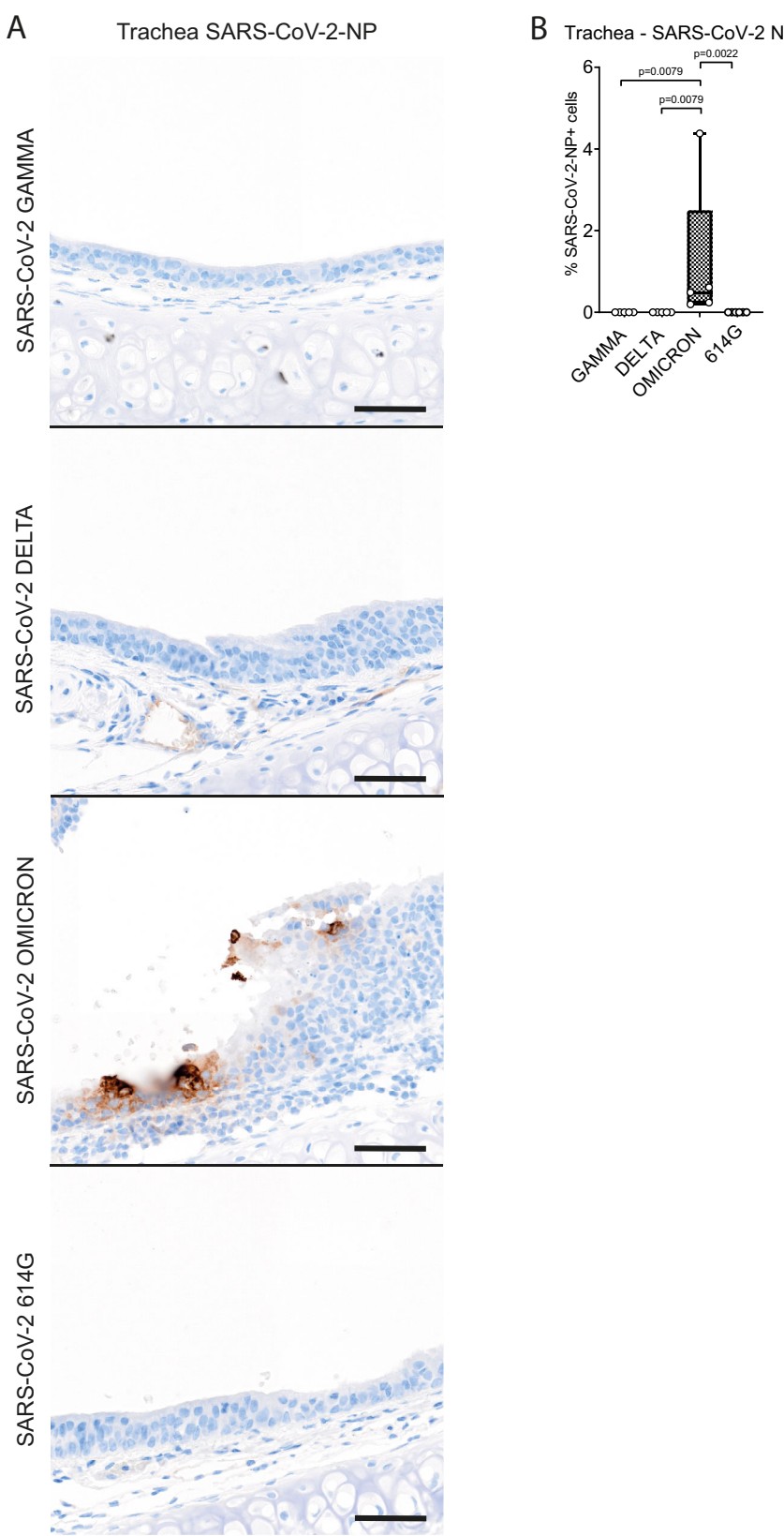

cells in the whole lung compared to VOCs Gamma and Delta. In the alveolar compartment, SARS-CoV-2 614 G and VOC Omicron infected hamsters revealed a significantly lower number of MPO+ cells compared to VOCs Delta and Gamma, or VOC Gamma infected hamsters, respectively. In the conductive airways, VOC Omicron infected hamsters had a significantly

lower number of MPO+ cells compared to VOCs Gamma and Delta infected hamsters. In the vasculature compartment, SARS-CoV-2 614 G showed a significantly lower number of MPO+ cells compared to VOCs Gamma and Delta infected hamsters. In addition, VOC Omicron infected hamsters showed a significantly lower number of IBA1+ cells in the whole lung compared to

**Fig. 5 SARS-CoV-2 NP viral antigen is still detectable after 4 dpi in the trachea of VOC Omicron infected hamsters. A** Representative images showing SARS-CoV-2 nucleoprotein (NP) immunolabeling in the trachea of hamsters infected with VOCs Gamma, Delta, Omicron or 614 G strain. Viral antigen (brown signal) was only detected in the tracheal epithelium of hamsters infected with VOC Omicron. Scale bars = 50 µm. **B** Quantification of SARS-CoV-2 NP antigen revealed a significantly higher number of immunolabeled cells in the trachea of hamsters infected with VOC Omicron compared to hamsters infected with VOCs Gamma, Delta or the 614 G strain. Data are shown as box and whisker plots. The bounds of the box plot indicate the 25th and 75th percentiles, the bar indicates medians, and the whiskers indicate minima and maxima. Data were tested by two-tailed Mann–Whitney-U tests followed by Benjamini–Hochberg correction. A p-value of ≤0.05 was chosen as the cut-off for statistical significance. N = 5 animals/group for VOCs Gamma and Delta and 6 animals/group for VOC Omicron and 614 G. For quantifications, one entire cross- and one entire longitudinal section were evaluated per animal. Source data are provided as a Source Data file.

VOCs Gamma and Delta. In the alveolar compartment, no statistically significant differences were observed among the groups, while in the conductive airways, VOC Omicron infected hamsters had a significantly lower number of IBA1$^+$ cells compared to VOC Gamma and SARS-CoV-2 614 G. In the vascular compartment, VOC Omicron showed a significantly lower number of IBA1$^+$ cells compared to VOCs Gamma and Delta. (Supplementary Table 4, Supplementary Fig. 3).

In summary, VOC Omicron infected hamsters showed a significantly decreased severity of lesions in the lower respiratory tract compared to the other investigated groups, with a notable sparing of alveoli. These findings led to the hypothesis that Omicron infects the lung to a lesser degree, as was also observed in the upper respiratory tract.

Viral antigen was observed in epithelial cells of the conductive airways and in type I and type II pneumocytes. The amount of SARS-CoV-2 NP antigen in the pulmonary conductive airways of VOC Omicron infected hamsters was significantly lower compared to VOC Gamma and SARS-CoV-2 614 G (Supplementary Table 4, Fig. 7). In addition, SARS-CoV-2 614 G infected animals showed significantly higher numbers of SARS-CoV-2 NP$^+$ cells compared to VOC Delta (Supplementary Table 4). The amount of viral antigen in the lung parenchyma, which included the alveolar and vascular compartment, was significantly lower in VOC Omicron infected hamsters when compared to VOCs Gamma and Delta infected ones (Supplementary Table 4, Fig. 7).

Infectious SARS-CoV-2 titers in the lung were the lowest in VOC Omicron infected hamsters, in line with the findings obtained by immunolabeling. The difference was statistically significant when compared to VOC Gamma and 614 G (Supplementary Table 4, Fig. 7). Interestingly, VOC Delta infected hamsters showed significantly lower SARS-CoV-2 titers in the lung than VOC Gamma and SARS-CoV-2 614 G infected hamsters (Supplementary Table 4).

Quantification of SARS-CoV-2 RNA in the lung confirmed the virus titration results. VOC Omicron infected hamsters showed the highest Ct values, corresponding to the lowest viral RNA copies. The Ct values were significantly higher compared to VOC Gamma. In addition, VOC Gamma infected hamsters showed significantly lower Ct values compared to 614 G and VOC Delta infected hamsters (Supplementary Table 4, Fig. 7).

Taken together, VOC Omicron infection in hamsters showed a decreased pathogenicity also for the lower respiratory tract and was characterized by a minimal to mild broncho-interstitial pneumonia with reduced SARS-CoV-2 NP antigen detection in the conductive airways and in the lung parenchyma, as well as decreased infectious SARS-CoV-2 titers and viral RNA loads. Collectively, these results demonstrate that VOC Omicron pathogenicity for the upper and lower respiratory tract is lower compared to SARS-CoV-2 614 G and VOCs Gamma and Delta.

**VOC Omicron infection in hamster does not spread to extra-respiratory organs.** The histopathological analysis of extra-

respiratory organs sampled during necropsy of VOC Omicron infected hamsters such as brain, liver, spleen, kidney, adrenal gland, stomach, small and large intestine, pancreas and testicles revealed no significant lesions attributable to SARS-CoV-2. Importantly, SARS-CoV-2 NP antigen was not detected in any tissue outside the respiratory tract. Similarly, infectious virus was not isolated from selected organs which represent frequent localizations of extra-respiratory SARS-CoV-2 detection (brain, intestine, kidney)[14,17].

Taken together, these results suggest that intranasal inoculation of hamsters with $10^4$ TCID$_{50}$ of SARS-CoV-2 VOC Omicron variant does not cause infection and histopathological lesions in extra-respiratory organs at 4 dpi.

## Discussion

There are clear indications that SARS-CoV-2 VOC Omicron is less pathogenic in humans and leads to fewer hospital submissions than previous SARS-CoV-2 variants[6], although limited information is available about the underlying mechanism. Therefore, animal models are used to compare the pathogenic potential of new emerging VOCs, such as VOC Omicron, with other previous variants. The pathogenicity of VOC Omicron has not been fully evaluated in the Syrian golden hamster model yet. In a previous study, infection with $10^3$ TCID$_{50}$ showed a lack of replication in the lung[18]. In our study, infection with $10^4$ TCID$_{50}$ did not cause weight loss in hamsters and showed a decreased pathogenicity for the upper and lower respiratory tract compared to VOCs Gamma and Delta and the SARS-CoV-2 614 G strain. In addition, viral antigen and RNA levels as well as infectious SARS-CoV-2 titers were also lower in the upper and lower respiratory tract of VOC Omicron infected animals.

Body weight loss is a well-established parameter to assess clinical severity of SARS-CoV-2 infection in the hamster model during the acute phase of the infection[15,19]. The VOC Omicron infected hamsters of the present study showed a mild weight increase of 3.1%, while in the other groups weight loss ranged from 6.5% for SARS-CoV-2 614 G to 11.8% and 10.8% for the VOCs Delta and Gamma, respectively. These results are in agreement with the reported body weight loss of ~10 to 15% in hamsters infected with a variety of SARS-CoV-2 variants[16]. Moreover, no weight loss occurred in hamsters infected with VOC Omicron[16]. Taken together, the results of the current study indicate a less severe clinical course of disease in VOC Omicron infected hamsters as compared to those infected with other SARS-CoV-2 strains.

While the replication of VOC Omicron and associated histo-pathology in the hamster lung have been demonstrated in a recent publication[16], histopathological changes and viral antigen distribution in the nasal turbinates have not been investigated. Hamsters infected with other SARS-CoV-2 strains and variants are reported to have a moderate to marked rhinitis at 4 dpi with histopathological lesions and viral antigen detection mainly centered on the olfactory mucosa. SARS-CoV-2 primarily targets the sustentacular cells in the olfactory mucosa, leading to a

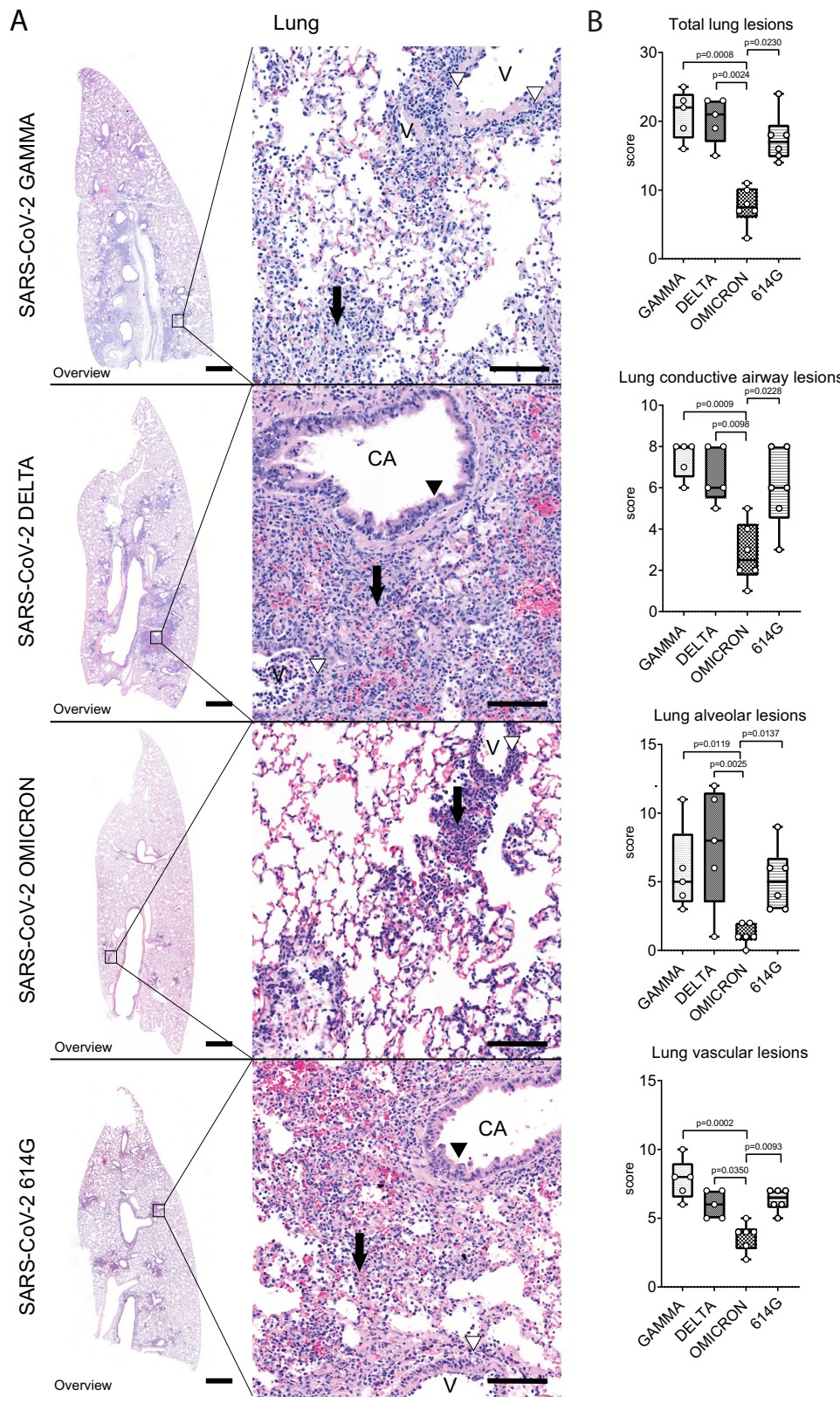

transient, massive necrosis and disintegration of the epithelium[20]. This phenomenon in hamsters is associated with the development of anosmia, a frequent clinical feature of COVID-19 patients[20–22]. In addition, it is also reported in some patients and in hamsters that the development of prolonged loss of smell could be caused by virus persistence in the olfactory mucosa[23]. In our study,

infection with SARS-CoV-2 614 G and VOCs Gamma and Delta also caused a widespread infection and damage in the olfactory epithelium, which agrees with previous reports. Interestingly, VOC Omicron infection in hamsters caused only a mild rhinitis with a largely intact olfactory epithelium. The significantly reduced epithelial damage was also associated with decreased

**Fig. 6 Decreased pathogenicity of VOC Omicron in the lungs compared to other SARS-CoV-2 strains. A** Representative images showing histopathological lesions in hamsters infected with VOCs Gamma, Delta, Omicron or 614 G strain. The left panel shows an overview of the left lung lobe and the right panel higher magnifications of lesions in the conductive airways (CA), vessels (V) and surrounding alveoli. Lungs of hamsters infected with VOCs Gamma or Delta or the 614 G strain showed the most extensive histopathological changes, which can be appreciated best on the overview pictures as consolidated, darker staining areas. Alveolar lesions were characterized by septal and luminal infiltration of macrophages and neutrophils/heterophils admixed with extravasated erythrocytes and fibrin, which obscured the alveolar architecture (arrows). Conductive airways frequently showed epithelial hyperplasia (black arrowheads). Vascular lesions mainly consisted of histiocytic-neutrophilic/heterophilic perivascular and intramural infiltrates (white arrowheads). Lungs of Omicron infected animals were only mildly affected in all lung compartments. Hematoxylin and eosin stain. Scale bars: 1 mm (overview) and 50 μm (high magnification). **B** Semi-quantitative analysis of pulmonary histopathology revealed significantly lower values in the overall lung score, lung conductive airway score, lung alveoli score, and lung vascular compartment score in VOC Omicron infected hamsters compared to the other investigated groups. Data are shown as box and whisker plots. The bounds of the box plot indicate the 25th and 75th percentiles, the bar indicates medians, and the whiskers indicate minima and maxima. Data were tested by two-tailed Mann–Whitney-*U* tests followed by Benjamini–Hochberg correction. A *p*-value of ≤0.05 was chosen as the cut-off for statistical significance. $N = 5$ animals/group for VOCs Gamma and Delta and six animals/group for VOC Omicron and 614 G. For quantifications, one entire cross and one entire longitudinal section of the left lung lobe were evaluated. Source data are provided as a Source Data file.

intraluminal exudates and immune cell infiltrates in the olfactory mucosa. In addition, SARS-CoV-2 NP antigen was rarely detected in the olfactory mucosa of VOC Omicron infected hamsters. These findings, together with low amount of viral RNA and low titers of infectious SARS-CoV-2 isolated from the nasal turbinates indicate a decreased pathogenicity of VOC Omicron for the upper respiratory tract. Lower viral RNA loads and infectious SARS-CoV-2 titers isolated from the nasal turbinates of VOC Omicron infected hamsters were reported previously[16]. Altogether, these results tempt to speculate that VOC Omicron infection in hamsters results in a decreased viral shedding from the upper respiratory tract secretions. However, only transmission experiments can substantiate this hypothesis. Moreover, the decreased damage of the olfactory mucosa caused by VOC Omicron infection most likely lowers the risk of developing anosmia. Preliminary data from human patients indeed suggests that anosmia appears to be a less frequent manifestation of VOC Omicron infection than reported for other strains and variants[24–26]. Given that anosmia is one of the most reported symptoms after SARS-CoV-2 infection[27], a decrease of anosmia incidence might favor more asymptomatic and undetected courses of the disease.

Interestingly, like for the nasal turbinates, histopathological changes, immune cell infiltrates and viral antigen distribution in the trachea of VOC Omicron infected hamsters have not been investigated yet. We detected no significant differences among the groups regarding pathological changes and numbers of MPO+ and IBA1+ cells. In addition, the very low numbers of SARS-CoV-2 NP+ cells detected at 4 dpi might suggest that the trachea is most likely not one of the main targets of this new SARS-CoV-2 VOC. Nevertheless, ciliary loss in the hamster tracheal epithelium following SARS-CoV-2 infection has been described[28], but whether VOC Omicron causes more marked or milder ciliary loss is still unknown. Future studies aiming at quantifying ciliary loss in VOC Omicron infected hamsters are warranted to shed light on this unanswered question. In contrast to the VOC Omicron group, no SARS-CoV-2 immunolabeled cells have been detected in the trachea of hamsters infected with other virus strains in this study. Tracheal infection by non-Omicron strains has been reported by others at 4 dpi[23,29,30] by qRT-PCR or viral titration. In particular, a recent study showed highest infectious virus titers in trachea at 2 dpi and then lower levels in most animals at 4 dpi[30]. Another recent study, using SARS-CoV-2 NP immunolabeling found high numbers of SARS-CoV-2+ cells at 1 dpi, but only few immunolabelled cells at 3 dpi[31]. These findings are in agreement with previous studies that reported very low numbers of positive cells[28,32] or positive signal limited to detached cell debris[33] at 3–4 dpi with non-Omicron strains. Therefore, we

assume that the trachea most likely was infected by non-Omicron strains, but that virus was largely or completely cleared below the detection level of immunolabeling at 4 dpi.

In contrast to nasal turbinates and trachea, lung histopathology in SARS-CoV-2 infected hamsters has been extensively described[22,34–37]. In addition, recent preliminary data on VOC Omicron infected hamsters also provided insights about lung lesions severity[16,38]. However, the reports lacked detailed quantification of the lesion severity based on a thorough scoring system of distinct pulmonary anatomical compartments. The data of the current study are in agreement with the aforementioned studies and complement data by providing additional quantification of immune cell infiltrates during the acute disease phase. Interestingly, our data show that VOC Omicron-induced lung histopathological lesions and inflammatory infiltrates were overall of lesser severity and extent than those in animals infected with SARS-CoV-2 614 G and the VOCs Gamma and Delta, as recently described in a comparative study among SARS-CoV-2 variants, including VOC Omicron, in hamsters and mice[16]. Lung lesions were more centered in the conductive airways and vascular compartments, while there was a notable sparing of the alveoli. VOC Omicron infected animals of our study showed either a complete absence or only minimal lesions in alveoli. This limited alveolar damage appears to parallel the milder clinical disease course in human patients, with lower risk of hospitalization and fatal outcome. Nevertheless, the occurrence of long-COVID after VOC Omicron infection is still under debate[39]. Therefore, future long-term studies are warranted to confirm whether this decreased pathogenicity in the alveolar compartment may results in a decreased occurrence and milder manifestations of long-term sequelae post infection.

SARS-CoV-2 viral antigen detection in the lung of infected hamsters is reported to be present in the lung conductive airways and alveolar cells until around 5 dpi[22], with a complete viral clearance around 7 dpi[22]. The current study reported a variable amount of SARS-CoV-2 NP antigen in the lung conductive airways and alveolar cells at 4 dpi with all the investigated strains. VOC Omicron infected animals also showed a lower number of immunolabeled cells in all pulmonary compartments and the lowest amount of viral RNA and infectious SARS-CoV-2 titers among all investigated groups. These lower lung viral titers are largely comparable to data obtained in an ex-vivo culture system of human lung tissue infected with multiple SARS-CoV-2 variants, including VOC Omicron[40]. Moreover, our results are in agreement with a comparative study among SARS-CoV-2 variants, including VOC Omicron, in hamsters and mice[16]. Altogether, these results show a decreased pathogenicity of VOC Omicron for the lower respiratory tract of hamsters.

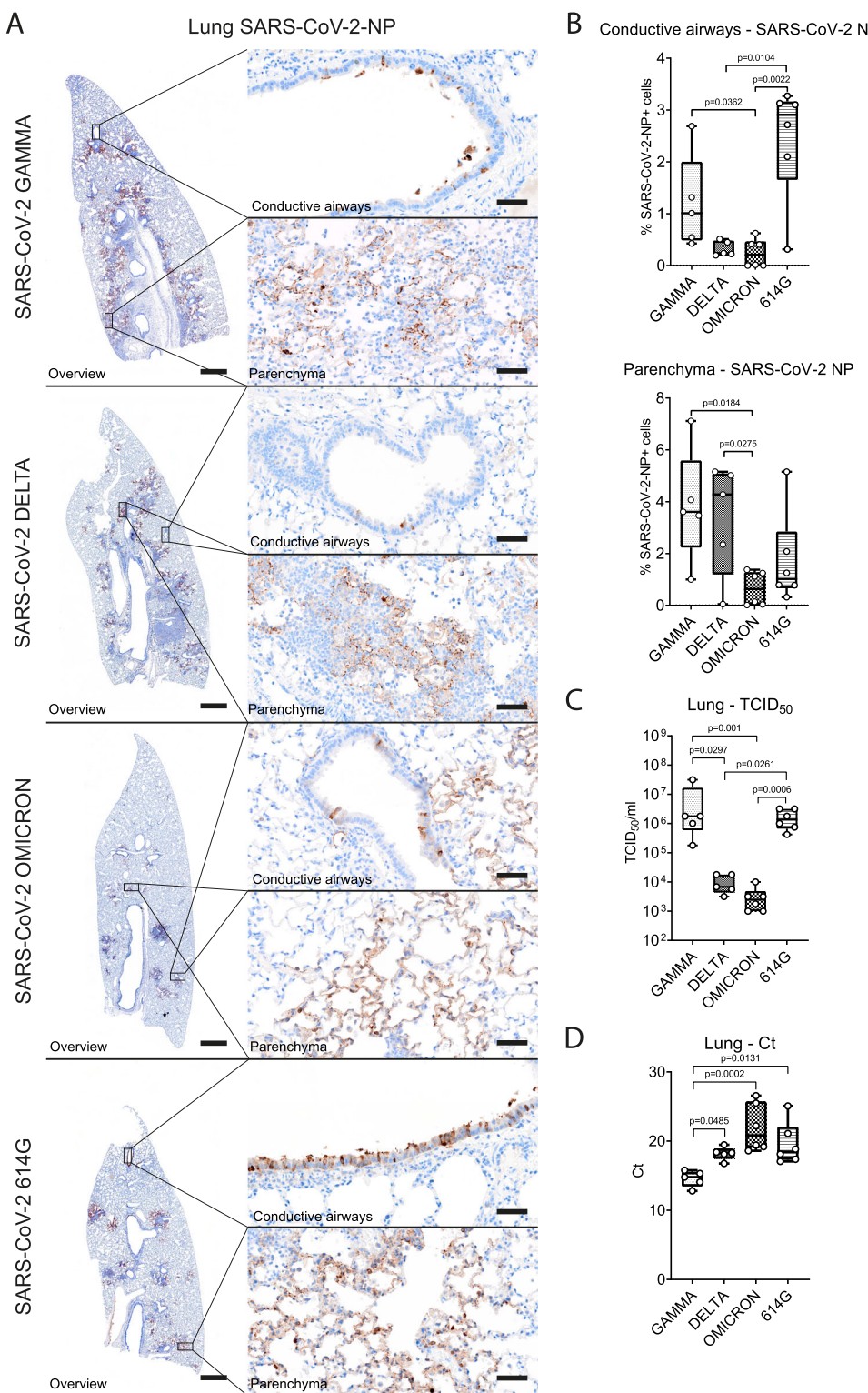

Based on these data, the question arises, how VOC Omicron can transmit more efficiently between humans. One may postulate that it is due to an increased virus replication of VOC Omicron especially in human airways, as also indicated by a study demonstrating an increased replication and virus release of VOC Omicron in ex vivo cultures of the human bronchus[40]. However, this has not been unequivocally reproduced in a rodent in vivo model yet. In our VOC Omicron infected hamsters, a higher viral load was detected by immunolabeling only in the trachea. We do not believe, however, that differences in transmissibility among VOC Omicron and other strains are related to this finding, since the overall number of SARS-CoV-2 positive cells in the organ was relatively low (maximal value below 5%, median below 1%). In contrast to the trachea, the viral load in both nasal cavity and lung as measured by immunohistochemistry, virus titration and qRT-PCR, was consistently lower in VOC Omicron infected hamsters. Similar findings were also reported by Halfmann et al. for hamsters and mice infected with VOC Omicron[16], although another study demonstrated higher numbers of SARS-CoV-2 NP immunolabeled epithelial cells in bronchioles close to the hilum of VOC

**Fig. 7 VOC Omicron infected hamsters showed decreased viral antigen, RNA and infectious viral titers in the lung compared to other SARS-CoV-2 strains. A** Representative images showing SARS-CoV-2 nucleoprotein (NP) immunolabeling in lungs of hamsters infected with VOCs Gamma, Delta, Omicron or 614 G strain. The left panel shows an overview of the left lung lobe and the right panel higher magnifications of viral antigen (brown signal) in the conductive airway epithelium and alveoli within the lung parenchyma. All SARS-CoV-2 infected hamsters showed immunolabeled cells, but the distribution and the amount varied among the groups. Scale bars: 1 mm (overview) and 50 μm (high magnification). **B** Quantification of SARS-CoV-2 NP antigen revealed significantly lower numbers of immunolabeled cells in the lung conductive airways of VOC Omicron infected hamsters compared to VOC Gamma and SARS-CoV-2 614 G infected ones. VOC Omicron infected hamsters showed also a significantly lower numbers of immunolabeled cells in the lung parenchyma compared to VOCs Gamma and Delta. **C** Infectious SARS-CoV-2 titers in the lung were significantly lower in VOC Omicron infected hamsters compared to VOC Gamma and SARS-CoV-2 614 G infected ones. **D** Quantification of SARS-CoV-2 RNA by qRT-PCR (Ct values). VOC Omicron infected hamsters show significantly higher Ct values, corresponding to low RNA content, compared to VOC Gamma infected hamsters. **B–D** Data are shown as box and whisker plots. The bounds of the box plot indicate the 25th and 75th percentiles, the bar indicates medians, and the whiskers indicate minima and maxima. Data were tested by two-tailed Mann–Whitney-*U* tests followed by Benjamini–Hochberg correction. A *p*-value of ≤0.05 was chosen as the cut-off for statistical significance. *N* = 5 animals/group for VOCs Gamma and Delta and 6 animals/group for VOC Omicron and 614 G. For quantifications, one entire longitudinal section of the left lung lobe was evaluated. Source data are provided as a Source Data file. Ct = cycle threshold; TCID$_{50}$ = tissue culture infection dose 50.

Omicron infected hamsters[32]. This did however not result in increased virus detection in oral swabs. Therefore, the relationship between increased viral antigen detection in the airways and increased viral shedding in VOC Omicron infected hamsters remains elusive. More studies are needed to come to a final conclusion, including transmission experiments and in vitro approaches with airway cell cultures from rodents. Additionally, other factors have to be taken into consideration as potential enhancers of VOC Omicron transmissibility, e.g., a higher affinity to ACE2, differences in cell entry mechanisms, stability of viral particles in the environment and the enhanced immune evasion of the variant[40–44].

SARS-CoV-2 infection of the CNS has been demonstrated in COVID-19 patients, murine models and in vitro in human brain organoids[45]. Neuroinvasion through the olfactory route has also been demonstrated in hamsters, which was associated with neuroinflammation in the olfactory bulb[23]. In the current study, neither viral antigen nor associated neuropathology or infectious virus were detected in the brain of VOC Omicron infected hamsters, probably due to the relatively limited infection observed in the olfactory mucosa. SARS-CoV-2 has also been reported to spread to extra-respiratory organs such as heart and kidney[46] or testes[47] in hamsters. Our hamsters did not show evidence of infection of extra-respiratory organs by histopathology, antigen detection and virus isolation, which may be due to the single time-point chosen for terminating the experiment.

The authors recognize that the study presents some limitations. First, virological and histological studies have been carried out at one time-point only. This does not allow to draw conclusions on the kinetics of VOC Omicron infection in comparison to other strains and variants as well as to assess, whether the observed milder lesions in VOC Omicron infected hamsters are not the result of slower dynamics, rather than decreased pathogenicity. However, since recent studies reported that lung pathogenicity at 6 dpi[16] or 7 dpi[32] was significantly lower in VOC Omicron infected hamsters when compared to other VOCs, we believe that VOC Omicron-induced lung pathogenicity in our study is truly lower and not biased by the chosen time-point. The choice of 4 dpi as an endpoint for the comparison was based on the typical time course of SARS-CoV-2 infection in the hamster using ancestral strains. Previous studies have shown that viral load in the respiratory tract peaks at 2 dpi[22]. However, the peak of lesion severity is observed between 3 and 6 dpi[19,34,36,37]. As the primary aim of this study was to characterize histological lesions caused by VOC Omicron in comparison with those of other variants, the time-point chosen represents a compromise between assessment of histopathology and viral load. Future studies involving flanking time-points are warranted in order to corroborate the conclusions

of the present study. Second, the experiment has been performed without a non-infected control group due to animal number limitations. One could argue that some of the observed mild lesions in the Omicron group could also be interpreted as background lesions or possible effects of intranasal inoculation. However, intralesional SARS-CoV-2 antigen was found in most foci with histopathological changes, suggesting an involvement of VOC Omicron.

Finally, it is important to note that we used 8–10 week old male hamsters. It was previously shown for the SARS-CoV-2 ancestral strain, that age- and sex-dependent differences in virus replication and shedding kinetics as well as lesion severity may exist[14,34,48,49]. To what extent these factors may influence these parameters for infections with the respective SARS-CoV-2 variants is subject of future studies.

In summary, the data presented confirm the lower pathogenicity of VOC Omicron compared to other SARS-CoV-2 strains in the hamster model and complement previous studies by providing a detailed analysis of histopathological lesions with additional quantification of immune cell infiltrates. Importantly, the study provides a thorough description of baseline changes at 4 dpi in hamsters infected with SARS-CoV-2 VOCs, including the unique histopathological investigation of the upper respiratory tract, which collectively will aid pre-clinical evaluation of SARS-CoV-2 prevention and treatment measures.

## Methods

**Hamster study**. Approval for the experiment including SARS-CoV-2 614 G Gamma and Delta variants was given by the Dutch authorities (Project license/ working protocol (WP) number: 27700202114492-WP12). The Omicron study was conducted in Hannover at the University of Veterinary Medicine, Foundation with the approval of the Niedersächsisches Landesamt für Verbraucherschutz und Lebensmittelsicherheit (LAVES file number 21/3755). The animals were under veterinary observation during the experiment and all efforts were made to minimize distress. Eight to ten weeks old male Syrian golden hamsters (*Mesocricetus auratus*) were purchased from Janvier Laboratories and were kept for 10 days under BSL-3 conditions prior to the experiment for acclimatization. Hamsters (*n* = 22) were divided in 4 groups of 5–6 hamsters each. They were infected by administrating a suspension containing $10^4$ TCID$_{50}$ with either, Gamma (*n* = 5), Delta (*n* = 5), Omicron (*n* = 6) VOCs or SARS-CoV-2 614 G (*n* = 6) respectively. For 4 days the animals were monitored and weighed twice daily until euthanized on day 4 post infection using an overdose of Ketamine and Medetomidin followed by exsanguination. Immediately after death, necropsies were performed, the right lung lobe and nasal swabs were collected and frozen at −70 °C for virological analyses. Subsequently, the left lung lobe, nasal turbinates, and tracheas were fixed in 10% buffered formalin (Chemie Vertrieb GmbH & Co Hannover KG, Hannover, Germany). In addition, brain, liver, spleen, kidney, adrenal gland, stomach, small and large intestine, pancreas and testicles from hamsters infected with the VOC Omicron were also collected. Lungs were pre-fixed by injections of 10% buffered formalin as recommended by Meyerholz and colleagues[50] to ensure an optimal histopathological evaluation. Nasal samples, following formalin fixation, were decalcified for about 14 days prior routine tissue processing.

**SARS-CoV-2 variants**. SARS-CoV-2 614 G (isolate Bavpat-1; European Virus Archive Global #026 V-03883) was grown to passage 3 on VeroE6 cells, and VOCs were grown to passage 3 on Calu-3 cells. For stock production, infections were performed at a multiplicity of infection (moi) of 0.01 and virus was collected at 72 h post infection, clarified by centrifugation and stored at −80 °C in aliquots. Stock titers were determined as described below. All work with infectious SARS-CoV-2 was performed in a Class II Biosafety Cabinet under BSL-3 conditions at Viroclinics Xplore.

Viral genome sequences were determined using Illumina deep-sequencing as described before[51]. The 614 G virus contained a spike S686G change in 48% of reads compared with the passage 1 (kindly provided by Dr. Christian Drosten) and no other variants >40%. The VOC Gamma, Delta and Omicron variant passage 3 sequences were identical to the original respiratory specimens and no minor variants >40% were detected. For VOC Omicron, the S1 region of spike was not covered well, due to primer mismatches. Therefore, the S1 region of the original respiratory specimen and passage 3 virus were confirmed to be identical by Sanger sequencing. VOC Gamma contained the following spike changes: L18F, T20N, P26S, D138Y, R190S, K417T, E484K, N501Y, D614G, H655Y, T1027I, V1176F. VOC Delta contained the following spike changes: T19R, G142D, del156–157, R158G, A222V, L452R, T478K, D614G, P681R and D950N. VOC Omicron contained the following spike mutations: A67VS, del69–70, T95I, G142-, del143–144, Y145D, del211, L212I, ins215EPE, G339D, S371L, S373P, S375F, K417N, N440K, G446S, S477N, T478K, E484A, Q493R, Q496S, Q498R, N501Y, Y505H, T547K, D614G, H655Y, N679K, P681H, N764K, D796Y, N856K, Q954H, N969K, L981F. The VOCs Gamma, Delta and Omicron sequences are available on Genbank under accession numbers OM442897, OM287123, and OM287553, respectively.

**Virus infectivity titration**. Organ samples were collected from nasal turbinates, lung, kidney, brain, small intestine. Virus infectious titers were determined in Vero cells for the SARS-CoV-2 614G, VOCs Gamma and Delta variants. Due to the limited replication of VOC Omicron in Vero cells, this variant was propagated and titrated in Calu-3 cells. Cells were seeded in 96 well plate and incubated at 37 °C. Then 24 h after seeding, cell culture media was replaced by DMEM + 2% FBS in the case of Vero cells that were infected with 10 fold serial dilutions of lung or nasal turbinate homogenate tissue samples. Plates were further incubated in a humidified atmosphere at 37 °C, 5% CO$_2$. Five days after infection, cytopathic effect was evaluated In the case of VOC Omicron titration, culture media was replaced for MEM + 2% FBS, 5dpi cells were fixed and stained using rabbit anti-SARS-CoV-2 nucleocapsid antibody (Sinobiological, Peking, China-40588-T62; dilution: 1:1000). A goat anti-rabbit IgG conjugated with Alexa fluor 488 was used as the secondary antibody (Thermofisher, Waltham, Massachusetts, USA, A11008; dilution: 1:1000). Viral titers (TCID50/ml) were calculated using the "Spearman–Kärber method"[52]. Analysis were performed in quadruplicates.

**qRT-PCR**. Viral RNA was extracted from nasal turbinates and lung using QIAmp Viral extraction kit according to the manufacturer's instructions. RT-qPCR assay was performed using the protocol established by the Institut Pasteur[53]. In brief, primers and probe targeting SARS-CoV-2 E gene were used following the Super Script III Platinum One-Step RT-qPCR (Invitrogen) protocol. Amplification was performed as followed: reverse transcription 55 °C 20 min, denaturation 95 °C 3 min, followed by 50x cycles of amplification at 95 °C 15 s, 58 °C 30 s, where data was acquired. Further analysis and Cq values were determined using the Bio-Rad CFX Maestro software (BioRad). Analysis was performed in quadruplicates.

**Histopathology**. Formalin-fixed paraffin embedded samples were cut into 2 μm thick serial sections and stained with hematoxylin and eosin (H&E). Sections of the nasal turbinates, trachea, and lung were scanned using an Olympus VS200 Digital slide scanner (Olympus Deutschland GmbH, Hamburg, Germany) and evaluated in a blinded manner with a semi-quantitative scoring system with special emphasis on inflammation, degeneration and regeneration as previously described, with minor modification[15]. Histopathological semi-quantitative evaluations were performed by veterinary pathologists. In particular, nasal turbinates, trachea and lung slides were evaluated in a blinded fashion and scored by FA and GB. Extra-respiratory organs from VOC Omicron infected hamsters were evaluated by LA. Subsequently, histopathological evaluation and scoring were reviewed and confirmed by board certified veterinary pathologists (WB, MC). Nasal turbinates were evaluated on a full length longitudinal section of the nose including respiratory and olfactory epithelium. Trachea was evaluated on cross- and longitudinal sections along the entire length of the organ. Finally, the lung was evaluated on one cross section (at the level of the entry of the main bronchus) and one longitudinal section (along the main bronchus) of the entire left lung lobe. The applied scoring systems are provided in details in Supplementary Table 5.

**Immunohistochemistry**. Immunohistochemistry of SARS-CoV-2 NP was performed using the Dako EnVision+ polymer system (Dako Agilent Pathology Solutions) and 3,3′-Diaminobenzidine tetrahydrochloride (Sigma-Aldrich, St. Louis, MO, United States) as previously described[36,37]. Monoclonal mouse primary antibody against SARS-CoV-2 NP (Sino Biological, Peking, China-40143-MM05;

dilution 1:16000) was applied overnight at 4 °C. In order to immunolabel macrophages/histiocytes and heterophils/neutrophils, immunohistochemistry of Ionized calcium-binding adapter molecule 1 (IBA1) and myeloperoxidase (MPO), respectively, was performed. The reaction was carried out using avidin–biotin complex (ABC) peroxidase kit (Vector Labs, Burlingame, CA, United States) and 3,3′-Diaminobenzidine tetrahydrochloride (Sigma-Aldrich, St. Louis, MO, United States) as previously described[36,54]. Rabbit polyclonal primary antibodies against IBA1 (FUJIFILM Wako Pure Chemical Corporation, Neuss, Germany; 019–19741; dilution 1:500) and MPO (Abcam, Cambridge, UK; ab9535; dilution 1:200) was applied overnight at 4 °C. A goat anti rabbit biotinylated IgG antibody, (Vector Labs, BA-1000; dilution: 1:200) was used as a secondary antibody. For negative controls, specific primary antibodies were replaced by ascitic fluid from non-immunized BALB/cJ mice (for SARS-CoV-2 NP), and serum from non-immunized rabbits (for IBA1 and MPO). The dilution of negative controls was chosen according to the protein concentration of replaced primary antibodies.

**Digital image analysis**. For the quantification of immunolabeled cells in nasal turbinates as well as in tracheal and pulmonary tissue, slides were digitized using the Olympus VS200 (Olympus Deutschland GmbH, Hamburg, Germany) slide scanner. Image analysis was performed using the open source software package QuPath version 0.3.1 for digital pathology image analysis[55]. For all animals and all immunolabelings, nasal turbinates and tracheal whole slides images as well as one longitudinal section (along the main bronchus) of the entire left lung lobe were evaluated. In brief, regions of interest (ROI), in the nasal turbinates (respiratory and olfactory mucosa) and the trachea (tracheal epithelum and subepithelial layer) were indicated by a veterinary pathologist. In particular for SARS-CoV-2 NP analysis in the lung, total lung tissue was detected automatically through digital thresholding and additional ROIs for conductive airways, including bronchi, bronchioles and terminal bronchioles were subsequently indicated by a veterinary pathologist. Lung parenchyma (alveolar and vascular compartments) was then obtained by subtraction of conductive airways from total lung tissue. The total numbers of immunolabeled and non-labeled cells was determined by automated cell detection in all ROIs, based on marker and tissue specific thresholding. For IBA1 and MPO analysis in the lung, total lung tissue was detected automatically through digital thresholding and additional ROIs for conductive airways, including bronchi, bronchioles and terminal bronchioles as well as vasculature were subsequently indicated by a veterinary pathologist. Alveoli were obtained by subtraction of conductive airways and vasculature from total lung tissue. The total numbers of immunolabeled and non-labeled cells were determined by automated cell detection in all ROIs, based on marker and tissue specific thresholding.

**Statistical analyses**. Statistical analyses and graph design were performed using GraphPad Prism 9.3.1 (GraphPad Software, San Diego, CA, USA) for Windows™. Data were tested for significant differences using Kruskal–Wallis tests. Pairwise comparisons among groups were obtained by two-tailed Mann–Whitney-$U$ tests and corrected for multiple group comparisons using the Benjamini–Hochberg correction. Statistical significance was accepted at exact $p$-values of ≤0.05.

**Reporting summary**. Further information on research design is available in the Nature Research Reporting Summary linked to this article.

## Data availability
Source data are provided with this paper.

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

## Acknowledgements
We thank Margarethe Jentzsch, Kristin Laudeley, Caroline Schütz, Julia Baskas, Kerstin Rohn, Kerstin Schöne and Jana Svea Harre for excellent technical support. This research was funded by the Deutsche Forschungsgemeinschaft (DFG; German Research Foundation, GB, LH, FK, WB and AO) -398066876/GRK 2485/1; BMBF (Federal Ministry of Education and Research) project entitled RAPID (Risk assessment in re-pandemic respiratory infectious diseases, GB, WB), 01KI1723G, Ministry of Science and Culture of Lower Saxony in Germany (14 – 76103–184 CORONA-15/20, WB and AO) and the EU SC1-PHE-CORONAVIRUS-2020 MANCO, no 101003651, AO). The study was also supported by the COVID-19 Research Network of the State of Lower Saxony (COFONI) with funding from the ministry of science and culture of Lower Saxony, Germany (14–76403–184, FA, MC, WB). This Open Access publication was funded by the Deutsche Forschungsgemeinschaft (DFG, German Research Foundation) - 491094227 "Open Access Publication Costs" and the University of Veterinary Medicine Hannover, Foundation.

## Author contributions

The study was designed by W.B., F.A., G.B., F.K., M.G.H., M.C., B.H., and A.O. Viral strains were propagated and sequenced by M.L., F.K., and M.G.H. Animal experiments were performed by G.v.A., K.G., F.K., and M.G.H. Pathology evaluation was performed by F.A., G.B., L.A., M.C., and W.B. Immunolabelling was conducted and analyzed by F.A., G.B., L.H., S.B., and M.R. Virus titration and PCR were performed and analyzed by F.K., M.G.H., and M.L. Data analysis and interpretation were performed by F.A., G.B., F.K., M.G.H., M.C., W.B., B.H., and A.O. Figures were prepared by F.A., G.B., L.A., and M.C. The original draft was written by F.A., G.B., L.A., W.B., M.C., F.K., and A.O. The manuscript was reviewed, edited, and approved by all authors. Funding was acquired by W.B. and A.O. The project was supervised by W.B., M.C., and A.O.

## Funding

## Competing interests

The authors declare no competing interests.
