## [Peer Review File · Nature Communications]

Reviewer comments ,first round –

Reviewer #1 (Remarks to the Author):

In this manuscript, Armando et al have investigated the Syrian golden hamster infection with the SARS-CoV-2 Omicron variant and associated pathology and compared it to hamsters infected with other Variants of concern. This study provides insight into the infection pattern and pathogenicity of the Omicron strain. The manuscript is generally well done and is in general agreement with prior studies. There are a few issues that should be addressed.

1. Line 78. Gross pathology of infected lungs is described for the Omicron and other variants. As these data are important and provide the basis for this study, they should be shown here.
2. Fig 2. The images appear to be of low resolution, making analysis difficult. Sections across variants should be matched for localization within the nasal epithelium to facilitate comparisons and the staining made uniform. Similarly, the higher power images should also be matched and their location on the low images on the left side indicated. Tissue sloughing should be indicated on both the low and high power images.
3. Fig 3. Similarly localized images of the infected olfactory mucosa should be shown. Presumably, sustentacular cells are infected by all of the variants and these cells should be indicated in the figure.
 - a. -How many individual mice and how many sections/mouse and areas within sections were analyzed?
4. Fig. 4- Ciliary loss is not obvious and should be indicated in the figure. In addition, a higher magnification figure may make cilia damage clearer.
5. Fig 5- Previous publications have shown that the trachea is infected by non-Omicron strains (de Melo et al., *Sci. Transl. Med.* 13, eabf8396 (2021), Plante, *J.A Nature* 592, 116–121 (2021) but the present study shows no infection in any strains other than Omicron. This disparity in results should be reconciled.
6. Line 202, figure 6- There is variability in the relative numbers of infected cells and virus titers, when hamsters infected with the various variants are compared. For example, infectious loads in the lungs were similar when Omicron and Delta variant hamsters were compared (even though differences are statistically significant) even though Omicron infected animals develop milder clinical disease. How many samples were analyzed? Do the authors have an explanation for these differences?
7. Line 207- Figure 7c is not called out.

Reviewer #2 (Remarks to the Author):

The authors have carried out a systematic virological, histological and immuno-histological comparison of different SARS-CoV-2 variants of concern (VOC) in upper and lower respiratory tract of Syrian golden hamsters. It is commendable that they have systematically investigated the pathology and immune-histology of the upper respiratory tract in the hamster model, often neglected in similar studies, which focus exclusively on the lungs. The authors conclude that VOC Omicron replicates less efficiently and causes less pathology in the upper and lower respiratory tract compared to the other viruses studied. The pathological observations correlate with differences in weight loss/gain which is the assessable parameter of clinical impact of these viruses with Omicron having no discernible weight loss as contrasted with other viruses investigated. They also fail to detect dissemination of SARS-CoV in extra-respiratory tissues. Overall, these findings correlate with the epidemiological observations that VOC Omicron has less severe clinical outcomes compared to previous VOCs or the wild type virus from early in 2020.

An interesting question then arises - why Omicron appears to transmit more efficiently, even though viral infection levels and pathology in the upper and lower respiratory tract is lower than for other variants. This question is not addressed, even speculatively, in the discussion. The one

difference to lower replication levels of Omicron in the respiratory tract appears to be the trachea. Although the pathology in the trachea associated with Omicron does not differ with other viruses, Figure 5 and line 146/7 suggests that numbers of virus infected cells is significantly increased. Viral titres in the trachea do not appear to have been compared? Could it be that the increased transmission seen in humans is driven by increased infectious aerosols released as a consequence of increased virus replication in the conducting airways? There are similar results reported from ex vivo culture so the human bronchus where viral titres with Omicron was markedly and significantly higher (Hui et al Nature 2022 doi: 10.1038/s41586-022-04479-6. PMID: 35104836).

One limitation of the current study is that virological and histological studies have been carried out only at one time point, i.e. day 4. While this is understandable from the point of view of the numbers of hamsters that would need to be used to carry out comparison of multiple viruses at multiple time points, it has previously been reported that peak viral load is seen at 2 days post infection and that infectious viral load decreases by (Sia et al Nature 2020). This should be mentioned in the limitations of the study.

Specific comments:

Line 65: typo. Should be "on" not "upon"

Line 152-153 appears redundant. Lines 192/3. This sentence is confusing. What exactly are the authors trying to say? It may be relevant to say that the only statistically significant difference between other VOCs is seen in the % NP cells in the conductive airways where there are differences between Delta and Gamma, and 614G vs. Delta.

In figure S4, the conductive airways NP cells refers to what exactly? Is this the bronchiolar airways within the lung parenchyma as contrasted to the tracheal or bronchus?

Table S4. Significantly lower lung TCID50 with Omicron. This is comparable with the significantly lower viral titres seen with ex vivo cultures of the human lung lung with Omicron vs, Delta and wild type virus (Hui et al Nature 2022 doi: 10.1038/s41586-022-04479-6. PMID: 35104836).

Reviewer #3 (Remarks to the Author):

- What are the noteworthy results?

The omicron variant of SARS-CoV-2 induces milder pathological changes in the upper and lower respiratory tract in Syrian golden hamsters than variants Gamma, Delta, or CoV-2 614G in male hamsters evaluated 4 days post-inoculation.

- Will the work be of significance to the field and related fields? How does it compare to the established literature? If the work is not original, please provide relevant references.

The work will contribute to knowledge of the hamster model of SARS-CoV-2 infection to include in-depth reporting of histopathology associated with variants. It may help researchers by providing a baseline of expected changes at 4 DPI which could aid in developing treatments/preventative measures. Established literature has found similar results in this model but also often provides additional measures of virus measurements within tissues such as ISH or qRTPCR for viral RNA. Additionally, published reports often provide information on infection at multiple time points post-inoculation and compare outcomes in males versus females.

- Does the work support the conclusions and claims, or is additional evidence needed?

The work generally supports the conclusions however the lack of uninfected (ie mock-inoculated) controls limits ability to interpret the mild changes attributed to omicron. Procedural controls would enable the investigators to assess both baseline histology and any effect of intranasal inoculations on respiratory system independent of infection at a comparable time-point.

SARS-CoV-2 Nucleoprotein was measured along with virus infectivity (TCID50) to compare levels of infectious virus in the nasal cavity and lungs. Assessing viral RNA via ISH or qRTPCR could validate these conclusions.

The time-line of omicron disease progression may differ from the other variants, which could explain why it appears to be less severe at day 4 in this study that is focused on a single time point post-inoculation. In fact, the differences noted could be temporal rather than spatial.

Additional time points should be evaluated to substantiate conclusions similar to recent reports in

this model

- Are there any flaws in the data analysis, interpretation and conclusions? - Do these prohibit publication or require revision?

Statistical analysis relies on pair-wise comparisons via Mann-Whitney tests without correcting appropriately for multiple group comparisons.

The interpretation that there is no viral spread to extra-respiratory organs based solely on IHC is questionable given sensitivity limitations of IHC. Additionally, viral spread to other organs and development of lesions may occur at time points besides 4 days PI.

- Is the methodology sound? Does the work meet the expected standards in your field?

The semi-objective histologic scoring system is thorough and meets standards in the field however other groups have used more advanced and objective approaches to measure cellular immune responses in this model via digital image analysis.

- Is there enough detail provided in the methods for the work to be reproduced?

Yes

Questions/comments:

Are there any images of the gross pathology? Do the lesions described correspond to histologic findings? Gross findings in these animals may be more related to peri-mortem changes consistent with euthanasia rather than viral infection.

Line 152-153: "After confirming the limited pathogenicity for the trachea, the investigations on the lower respiratory tract were completed with the lung investigations" - awkwardly worded and unclear

Line 209 - which tissues were tested for NP antigen?

Line 380 - did all evaluators score all slides?

Figure 2 Are all arrowheads meant to indicate intact olfactory epithelium? Confusing for images in Gamma and Delta. Also the labeling in the legend is not consistent with the image.

Figure 3 legend labeling not consistent with image

Figure 6 legend line 615: "lungs of omicron infected animals were mildly affected with a marked affected and not in all lung compartments" - what is meant by this?

Scoring of trachea describes bronchi/bronchioles

REVIEWER COMMENTS

Reviewer #1 (Remarks to the Author):

1) *"In this manuscript, Armando et al have investigated the Syrian golden hamster infection with the SARS-CoV-2 Omicron variant and associated pathology and compared it to hamsters infected with other Variants of concern. This study provides insight into the infection pattern and pathogenicity of the Omicron strain. The manuscript is generally well done and is in general agreement with prior studies. There are a few issues that should be addressed."*

We thank Reviewer 1 for the consideration. We have adapted the manuscript according to the valuable suggestions, which has improved the scientific value and presentation of the work. We hope that the current version of the manuscript is acceptable for publication in its current form.

2) *"1. Line 78. Gross pathology of infected lungs is described for the Omicron and other variants. As these data are important and provide the basis for this study, they should be shown here."*

We agree with Reviewer 1 that representative gross pathology pictures of the lungs from the 4 investigated groups might help to understand the findings commented in line 78 and to visualize the differences among the groups. Unfortunately, gross pictures have not been taken of all animals, in part for logistic reasons. Retrospectively, this is a pity and would have improved data presentation. However, the authors would like to point out that the presentation of the histological lung lesions in the manuscript includes sub-gross pictures of the entire left lung lobe section (Figures 6 and 7), which gives the reader an overview of the differences in severity and distribution pattern of the lesions among the groups. The authors feel that the histopathological sub-gross pictures very specifically show virus-induced lesions similar to, and probably more detailed than, gross pathology images.

3) *"2. Fig 2. The images appear to be of low resolution, making analysis difficult."*

We sincerely apologize with Reviewer 1 for the shortcoming, now Fig.2 has been re-uploaded with high resolution (300 DPI, see response to point 4).

4) *"Sections across variants should be matched for localization within the nasal epithelium to facilitate comparisons and the staining made uniform. Similarly, the higher power images should also be matched and their location on the low images on the left side indicated. Tissue sloughing should be indicated on both the low and high power images"*

We thank Reviewer 1 for these kind suggestions that have improved the data presentation and allow the reader a better comparison between the groups. The overview pictures now show the same localization of the caudal nasal turbinates for each group and are taken at a lower magnification, which allows a better orientation. The

high magnification images are taken from slightly different localizations within the area shown at low magnification pictures (now indicated by rectangles), since lesion distribution in SARS-CoV-2 infection has a multifocal pattern and lesions are not always present in the exact same localizations. The high-power images are representative images showing type and severity of the lesions typically found within the respective group. According to the Reviewer's suggestion, tissue sloughing and intraluminal debris has been indicated by asterisks in the low and high power images. Additionally, individual cell death is now indicated by arrows in the high magnification images. We feel that indeed these changes improve the clarity of the figure and give the reader a better impression of histopathological lesions.

The following changes have been made:

FIGURES

- now Figure 2 looks as follows:

Figure 2: Decreased pathogenicity of VOC Omicron for nasal olfactory mucosa compared to other SARS-CoV-2 strains.

A) Representative images showing an overview of the caudal nasal turbinates (left panel) and high magnification of the olfactory mucosa (right panel) of hamsters infected with VOCs Gamma, Delta, Omicron or 614G strain. Olfactory mucosa in VOCs Gamma and Delta or 614G infected hamsters showed the most striking histopathological changes. Lesions were mainly characterized by disorganization, cell death (arrows) and cell sloughing, associated with intraluminal exudate composed of proteinaceous fluid, cell debris and degenerated neutrophils/heterophils (asterisks). Olfactory mucosa in VOC Omicron infected hamsters was largely intact and only occasional cell death was observed (arrow). Scale bars = 1 mm (overviews) and 50 μ m (high magnifications). Hematoxylin and eosin stain. B) Semi-quantitative analysis of nasal turbinates histopathology revealed significantly lower values in the overall score as well as the separate scores for respiratory and olfactory mucosa in VOC Omicron infected hamsters. Data are shown as box and whisker plots with median and quartiles. Significant differences between the infection groups are indicated by * (* $p \leq 0.05$, ** $p \leq 0.01$, **** $p \leq 0.0001$).

Data was tested by pairwise Mann-Whitney-U tests followed by Benjamini-Hochberg correction. N= 5 animals/group for VOCs Gamma and Delta and 6 animals/group for VOC Omicron and 614G. For quantification, one entire longitudinal section of the nasal turbinates was evaluated per animal.

5) "3. Fig 3. Similarly localized images of the infected olfactory mucosa should be shown. "

The images and figure panel have been modified in a similar fashion as described above for Figure 2.

The following changes have been made:

FIGURES

- now figure 3 looks as follows:

Figure 3: VOC Omicron infected hamsters showed decreased viral antigen and infectious viral titers in the nasal turbinates compared to other SARS-CoV-2 strains.

A) Representative images showing SARS-CoV-2 nucleoprotein (NP) immunolabeling in the nasal turbinates of hamsters infected with VOCs Gamma, Delta, Omicron or 614G strain. The images depict an overview of the caudal nasal turbinates (left panel) and high magnification of the olfactory mucosa (right panel). Viral antigen (brown signal) was mainly detected in epithelial cells within the olfactory mucosa and within intraluminal exudates (asterisks). Based on the cell morphology, characterized by apically located nuclei and abundant apical cytoplasm, most of these cells were sustentacular cells (arrowheads). The number of positive cells was highest in hamsters infected with 614 G, followed by hamsters infected with VOCs Gamma or Delta. In hamsters infected with VOC Omicron, only rare foci with a few positive cells were observed. Scale bars: 1 mm (overview) and 20 μ m (high magnification). B) Quantification of SARS-CoV-2 NP antigen in the respiratory and olfactory mucosa. VOC Omicron infected hamsters displayed a significantly lower number of immunolabeled cells in the olfactory mucosa compared to hamsters infected with 614G C) Quantification of infectious SARS-CoV-2 titers in the nasal turbinates. Titers were significantly lower in VOC Omicron infected hamsters compared to 614G and VOC Gamma infected hamsters. D) Quantification of SARS-CoV-2 RNA by qRT-PCR (Ct values). VOC Omicron infected hamsters show significantly higher Ct values, corresponding to low RNA content, compared to VOCs Gamma and Delta infected hamsters. B-D) Data are shown as box and whisker plots with median and quartiles. Significant differences between the infection groups are indicated by * (* $p \leq 0.05$, ** $p \leq 0.01$, *** $p \leq 0.001$, **** $p \leq 0.0001$). Data was tested by pairwise Mann-Whitney-U tests followed by Benjamini-Hochberg correction. N= 5 animals/group for VOCs Gamma and Delta and 6 animals/group for VOC Omicron and 614G. For quantifications, one entire longitudinal section of the nasal turbinates was evaluated per animal.

6) "Presumably, sustentacular cells are infected by all of the variants and these cells should be indicated in the figure."

We agree with this comment given by Reviewer 1. In the modified version of the figure panel, infected sustentacular cells are indicated by an arrowhead (see above in the response to point 5).

7) "a. -How many individual mice and how many sections/mouse and areas within sections were analyzed?"

We apologize for the lack of clarity. The number of animals was 5 per group for VOCs Gamma and Delta and 6 per group for VOC Omicron and 614G. Nasal turbinates were evaluated on a full length longitudinal section of the nose including respiratory and olfactory epithelium. Trachea was evaluated on one cross- and one longitudinal section along the entire length of the organ. The lung was evaluated on one cross section at the level of the entry of the main bronchus and one longitudinal section along the main bronchus of the left lung lobe. The entire sections were analyzed. To improve clarity, the information about the number of animals per group and the number of evaluated sections per animal has been included in the figure legends in the revised version of the manuscript.

The following changes have been made:

Figure 2 legend

-now lines 728-728 read as follows: "N= 5 animals/group for VOCs Gamma and Delta and 6 animals/group for VOC Omicron and 614G. For quantifications, one entire longitudinal section of the nasal turbinates was evaluated per animal."

Figure 3 legend

-now lines 750-752 read as follows: "N= 5 animals/group for VOCs Gamma and Delta and 6 animals/group for VOC Omicron and 614G. For quantifications, one entire longitudinal section of the nasal turbinates was evaluated per animal."

Figure 4 legend

-now lines 762-764 read as follows: "N= 5 animals/group for VOCs Gamma and Delta and 6 animals/group for VOC Omicron and 614G. For quantifications, one entire cross- and one entire longitudinal section were evaluated per animal."

Figure 5 legend

-now lines 775-777 read as follows: "N= 5 animals/group for VOCs Gamma and Delta and 6 animals/group for VOC Omicron and 614G. For quantifications, one entire cross- and one entire longitudinal section were evaluated per animal."

Figure 6 legend

-now lines 795-797 read as follows: "N= 5 animals/group for VOCs Gamma and Delta and 6 animals/group for VOC Omicron and 614G. For quantifications, one entire cross and one entire longitudinal section of the left lung lobe were evaluated. "

Figure 7 legend

-now lines 816-818 read as follows: "N= 5 animals/group for VOCs Gamma and Delta and 6 animals/group for VOC Omicron and 614G. For quantifications, one entire longitudinal section of the left lung lobe was evaluated."

8) " 4. Fig. 4-Ciliary loss is not obvious and should be indicated in the figure. In addition, a higher magnification figure may make cilia damage clearer."

We agree with the comment given by Reviewer 1. To improve visualization of the ciliary loss, higher magnification images have been included in the figure panel (Figure 4). In these images, the few remaining ciliated cells are indicated by arrowheads.

The following changes have been made:

FIGURES

- now Figure 4 looks as follows:

Figure 4: Tracheal lesions are mild to moderate regardless of the SARS-CoV-2 strain infection in hamsters.

A) Representative images showing tracheal lesions in hamsters infected with VOCs Gamma, Delta, Omicron or 614G strain. Trachea histopathological changes were characterized by neutrophilic/heterophilic exocytosis (arrows) and sub-epithelial infiltration of macrophages and neutrophils/heterophils (asterisks). The insets show a higher magnification of the ciliated epithelium. Intact cilia are indicated by arrowheads. In all groups, most cells show ciliary loss. Hematoxylin and eosin stain. Scale bars = 50 μ m and 10 μ m (inset). B). Semi-quantitative analysis of tracheal histopathology revealed no statistically significant differences in the recorded scores among the groups. Data are shown as box and whisker plots with median and quartiles. Data was tested by pairwise Mann-Whitney-U tests followed by Benjamini-Hochberg correction. N = 5 animals/group for VOCs Gamma and Delta and 6 animals/group for VOC Omicron and 614G. For quantifications, one entire cross- and one entire longitudinal section were evaluated per animal.

9) “ 5. Fig 5-Previous publications have shown that the trachea is infected by non-Omicron strains (de Melo et al., *Sci. Transl. Med.*13, eabf8396 (2021), Plante, *J.A Nature* 592, 116–121 (2021) but the present study shows no infection in any strains other than Omicron. This disparity in results should be reconciled.”

We agree with Reviewer 1 suggestion that the disparity should be addressed in the manuscript. Among others, De melo et al., 2021 (DOI: 10.1126/scitranslmed.abf8396) and Plante et al., 2021(DOI: 10.1038/s41586-020-2895-3) detected SARS-CoV-2 tracheal infection at 4 dpi in non-Omicron strains. However, the mentioned studies used qRT-PCR for virus detection, while our study used SARS-CoV-2 NP immunolabeling of tracheal tissues. The authors feel that the disparity in the results is rather due to a difference in sensibility of the two methods instead of a lack of infection of the trachea by the non-Omicron strains. Indeed, other studies using SARS-CoV-2 NP immunolabeling of tracheal tissues also show very low numbers of positive cells (Robinot et al., 2021, *Nature communications*, DOI: 10.1038/s41467-021-24521-x; Suzuki et al., 2022, *Nature*, DOI: 10.1038/s41586-022-04462-1) or positive signal limited to detached cell debris (Chan et al., 2020, *Clinical Infectious Diseases*, DOI: 10.1093/cid/ciaa325) at 3-4 dpi following infection with non-Omicron strains. Based on the authors’ findings in

a longitudinal study (1, 3, 6 and 14 dpi) using an ancestral SARS-CoV-2 strain, tracheal infection is very transient with abundant SARS-CoV-2 NP positive signal at 1 dpi, but largely absent signal at 3 dpi (unpublished data).

Therefore, regarding the submitted manuscript, the authors assume that the trachea most likely was infected by all non-Omicron strains at an earlier time-point, but that infection has been cleared or reduced to a level below the detection level of immunolabeling, which seems to be a typical course for most SARS-CoV-2 strains to date. Unfortunately, qRT-PCR of tracheal tissues cannot be performed for this study to detect the suspected lower levels of virus, since frozen tissues were not collected. The interesting finding is that viral antigen was detectable in VOC Omicron infected animals, albeit in a low number, which could point to a prolonged or delayed tracheal infection. The interpretation of the tracheal findings has been addressed in a new paragraph of the discussion in the revised version of the manuscript.

The following changes have been made:

DISCUSSION

-now lines 316-324 read as follows: “In contrast to the VOC Omicron group, no SARS-CoV-2 immunolabeled cells have been detected in the trachea of hamsters infected with other virus strains in this study. Tracheal infection by non-Omicron strains has been reported by others at 4 dpi^{24,30}. However, those studies used qRT-PCR for virus detection, instead of less sensitive SARS-CoV-2 NP immunolabeling. Other studies using SARS-CoV-2 NP immunolabeling show very low numbers of positive cells^{29,31} or positive signal limited to detached cell debris³² at 3-4 dpi with non-Omicron strains. Therefore, we assume that the trachea most likely was infected by non-Omicron strains, but that virus was largely or completely cleared below the detection level of immunolabeling at 4 dpi.”

10) “6. Line 202, figure 6-There is variability in the relative numbers of infected cells and virus titers, when hamsters infected with the various variants are compared. For example, infectious loads in the lungs were similar when Omicron and Delta variant hamsters were compared (even though differences are statistically significant) even though Omicron infected animals develop milder clinical disease. How many samples were analyzed? Do the authors have an explanation for these differences? “

We thank Reviewer 1 for pointing out this variability in viral load among the groups that is not identical in the two quantification methods. However, both showed a marked difference between VOC Omicron infected hamsters and VOC Gamma as well as 614G. Therefore, we are convinced that the encountered variability was not affecting the overall interpretation of our findings. However, to further corroborate the statement of lower viral loads in VOC Omicron infected animals, we additionally included results of qRT-PCR-based quantification in the revised version of the manuscript. This analysis also confirmed that viral load is markedly lower in VOC Omicron infected hamsters. For each method, one sample per animal was analyzed in quadruplicates. In all three methods, there is some degree of deviation regarding the relative amounts of detectable virus. To answer Reviewer 1 question regarding a possible explanation, we assume that these differences are largely due to various technical reasons. First, virus is not homogenously distributed within the lung and the number of viral particles can vary among samples processed for histology, virus titration and qRT-PCR. Secondly, there are differences in the coverage of processed tissue, e.g. in histology, a two-dimensional section at a certain level is analyzed, while a tissue homogenate is used for virus titration and qRT-PCR. Moreover, the methods have different sensitivities and targets (viral protein, RNA and infectious particles), which do not necessarily correlate.

The following changes have been made:

RESULTS

-now lines 140-144 read as follows: “Quantification of SARS-CoV-2 RNA in the nasal turbinates confirmed the virus titration results. VOC Omicron infected hamsters showed the highest Ct values, corresponding to the lowest viral RNA copies. The Ct values were significantly higher compared to VOC Delta and Gamma. In addition, VOC Gamma infected hamsters showed significantly lower Ct values compared to 614G infected hamsters (Supplementary Table S2, Figure 3).”

-now lines 235-239 read as follows: “Quantification of SARS-CoV-2 RNA in the lung confirmed the virus titration results. VOC Omicron infected hamsters showed the highest Ct values, corresponding to the lowest viral RNA copies. The Ct values were significantly higher compared to VOC Gamma. In addition, VOC Gamma infected hamsters showed significantly lower Ct values compared to 614G and VOC Delta infected hamsters (Supplementary Table S4, Figure 7).”

MATERIALS AND METHODS

-now lines 468-475 read as follows:

“qRT-PCR

Viral RNA was extracted from nasal turbinates and lung using QIAmp Viral extraction kit according to the manufacturer’s instructions. RT-qPCR assay was performed using the protocol established by the Institut Pasteur⁵². In brief, primers and probe targeting SARS-CoV-2 E gene were used following the Super Script III Platinum One-Step RT-qPCR (Invitrogen) protocol. Amplification was performed as followed: reverse transcription 55 °C 20 min, denaturation 95 °C 3 min, followed by 50x cycles of amplification at 95 °C 15 s, 58 °C 30 s, where data was acquired. Further analysis and Cq values were determined using the Bio-Rad CFX Maestro software (BioRad). Analysis was performed in quadruplicates.”

MATERIALS AND METHODS

-now lines 457-467 read as follows (changes are underlined):

“Virus infectivity titration

Organ samples were collected from nasal turbinates, lung, kidney, brain, small intestine. Virus infectious titers were determined in Vero cells for the SARS-CoV-2 614G, VOCs Gamma and Delta variants. Due to the limited replication of VOC Omicron in Vero cells, this variant was propagated and titrated in Calu-3 cells. Cells were seeded in 96 well plate and incubated at 37°C. Then 24h after seeding, cell culture media was replaced by DMEM + 2% FBS in the case of Vero cells that were infected with 10 fold serial dilutions of lung or nasal turbinate homogenate tissue samples. Plates were further incubated in a humidified atmosphere at 37°C, 5 % CO₂. Five days after infection, cytopathic effect was evaluated In the case of VOC Omicron titration, culture media was replaced for MEM + 2% FBS, 5dpi cells were fixed and stained using anti-SARS-CoV-2 Nucleocapsid antibody (Sinobiological). Viral titers (TCID₅₀/ml) were calculated using the “Spearman-Kärber method” 40. Analysis were performed in quadruplicates.”

11) “7. Line 207-Figure 7c is not called out.”

We sincerely apologize to Reviewer 1 for the shortcoming, Figure 7 is now mentioned in the respective results section.

Reviewer #2 (Remarks to the Author):

1) “The authors have carried out a systematic virological, histological and immuno-histological comparison of different SARS-CoV-2 variants of concern (VOC) in upper and lower respiratory tract of Syrian golden hamsters. It is commendable that they have systematically investigated the pathology and immune-histology of the upper respiratory tract in the hamster model, often neglected in similar studies, which focus exclusively on the lungs. The authors conclude that VOC Omicron replicates less efficiently and causes less pathology in the upper and lower respiratory tract compared to the other viruses studied. The pathological observation correlate with differences in weight loss/gain which is the assessable parameter of clinical impact of these viruses with Omicron having no discernible weight loss as contrasted with other viruses investigated. They also fail to detect dissemination of SARS-CoV in extra-respiratory tissues. Overall, these findings correlate with the epidemiological observations that VOC Omicron has less severe clinical outcomes compared to previous VOCs or the wild type virus from early in 2020.”

We thank Reviewer 2 for the appreciations and we hope that the current version of the manuscript can be positively evaluated after adaptations based on valuable suggestions that will improve its scientific contents.

2) “An interesting question then arises - why Omicron appears to transmit more efficiently, even though viral infection levels and pathology in the upper and lower respiratory tract is lower than for other variants. This question is not addressed, even speculatively, in the discussion. The one difference to lower replication levels of Omicron in the respiratory tract appears to be the trachea. Although the pathology in the trachea associated with Omicron does not differ with other viruses, Figure 5 and line 146/7 suggests that numbers of virus infected cells is significantly increased. Viral titres in the trachea do not appear to have been compared? Could it be that the increased transmission seen in humans is driven by increased infectious aerosols released as a consequence of increased virus replication in the conducting airways? There are similar results reported from ex vivo culture so the human bronchus where viral titres with Omicron was markedly and significantly higher (Hui et al Nature 2022 doi: [10.1038/s41586-022-04479-6](https://doi.org/10.1038/s41586-022-04479-6). PMID: 35104836).”

We thank Reviewer 2 for pointing out this very intriguing and relevant point. Indeed, the study performed by Hui et al. elegantly demonstrates an increased replication and virus release of VOC Omicron in human airway epithelium *in vitro* Hui et al., 2022 (DOI: [10.1038/s41586-022-04479-6](https://doi.org/10.1038/s41586-022-04479-6)). It remains however unclear, whether this phenomenon also occurs in rodents *in vivo*, since it has not been unequivocally reproduced in a rodent *in vivo* model yet. In our study, the only location in which a higher viral load was detected by immunohistochemistry in VOC Omicron infected hamsters was the trachea. Unfortunately, viral titration or qRT-PCR of tracheal tissues could not be performed to substantiate the finding, since frozen tissues were not collected of the organ. However, the authors do not believe that the finding in the trachea can explain differences in transmissibility among VOC Omicron and other strains, since the number of SARS-CoV-2 positive cells in the organ was very low (maximal value below 5%, median below 1%), raising questions about its biological significance. In contrast to the trachea, the viral load in the nasal cavity and the lung was consistently lower in VOC Omicron infected hamsters compared to other strains, as demonstrated by immunolabeling, virus titration and qRT-PCR. In line with our findings, Halfmann et al. also reported lower levels of bronchial infection in hamsters and mice and lower amounts of virus in the nasal turbinates of mice following VOC Omicron infection, compared to other strains Halfmann et al., 2022 (DOI: [10.1038/s41586-022-04441-6](https://doi.org/10.1038/s41586-022-04441-6)). In contrast, another study that focused the analysis on the bronchioles that are included in the lung area close to the hilum demonstrated higher numbers of SARS-CoV-2 NP immunopositive cells of hamsters infected with VOC Omicron compared to VOC Delta and B.1.1. at 3 dpi Suzuki et al., 2022 (DOI: [10.1038/s41586-022-04462-1](https://doi.org/10.1038/s41586-022-04462-1)). However, this difference did not result in increased virus detection in oral swabs collected on 7 consecutive days following infection. In conclusion, it remains elusive whether an increased viral antigen detection in the airways results in an increased viral shedding in VOC Omicron infected hamsters. More studies are needed for a final interpretation, optimally including

transmission experiments in rodents as well as complementary in vitro approaches using airway cell cultures from the same species.

Besides the magnitude of virus release by the index host, other intrinsic factors must be taken into consideration as potential enhancers of VOC Omicron transmissibility such as an enhanced cell tropism and infectivity. For instance, the Omicron variant bears more than 30 mutations that influence the affinity to ACE2 receptor and furin-mediated spike cleavage Araf et al., 2022 (DOI: 10.1002/jmv.27588); Garcia-Beltran et al., 2022 (DOI: 10.1016/j.cell.2021.12.033). Moreover, it has also been demonstrated that VOC Omicron efficiently uses a TMPRSS2-independent, endocytic cell entry pathway in contrast to other strains, which could have an impact on cell tropism and infectivity Hui et al., 2022 (DOI: 10.1038/s41586-022-04479-6); Zhao et al., 2022 (DOI: 10.1080/22221751.2021.2023329). Recent evidence also suggests a higher stability of VOC Omicron virus particles in the environment Chin et al., 2022 (DOI: 10.1101/2022.03.09.483703 biorxiv). Finally, VOC Omicron shows the highest rate of antibody evasion among all SARS-CoV-2 isolates, which certainly facilitates infection of the individual and contributes to a fast spread within the population Ju et al., 2022 (DOI: 10.1038/s41422-022-00638-6). More extensive future studies are needed to elucidate which factors play together to increase transmissibility of VOC Omicron, which also would have important implications for the risk assessment of other VOCs in the future.

In order to address this highly interesting issue in the manuscript, an additional paragraph has been added to the discussion.

The following changes have been made:

DISCUSSION

-now lines 354-374 read as follows: “Based on these data, the question arises, how VOC Omicron can transmit more efficiently between humans. One may postulate that it is due to an increased virus replication of VOC Omicron especially in human airways, as also indicated by a study demonstrating an increased replication and virus release of VOC Omicron in ex vivo cultures of the human bronchus³⁹. However, this has not been unequivocally reproduced in a rodent in vivo model yet. In our VOC Omicron infected hamsters, a higher viral load was detected by immunolabeling only in the trachea. We do not believe however, that differences in transmissibility among VOC Omicron and other strains is related to this finding, since the overall number of SARS-CoV-2 positive cells in the organ was relatively low (maximal value below 5%, median below 1%). In contrast to the trachea, the viral load in both nasal cavity and lung as measured by immunohistochemistry, virus titration and qRT-PCR, was consistently lower in VOC Omicron infected hamsters. Silmilar findings were also reported by Halfmann et al. for hamsters and mice infected with VOC Omicron¹⁷, although another study demonstrated higher numbers of SARS-CoV-2 NP immunolabelled epithelial cells in bronchioles close to the hilum of VOC Omicron infected hamsters.³¹ This did however not result in increased virus detection in oral swabs. Therefore, the relationship between increased viral antigen detection in the airways and increased viral shedding in VOC Omicron infected hamsters remains elusive. Therefore, more studies are needed to come to a final conclusion, including transmission experiments and in vitro approaches with airway cell cultures from rodents. Additionally, other factors have to be taken into consideration as potential enhancers of VOC Omicron transmissibility, e.g. a higher affinity to ACE2, differences in cell entry mechanisms, stability of viral particles in the environment and the enhanced immune evasion of the variant³⁹⁻⁴³.”

3) “One limitation of the current study is that virological and histological studies have been carried out only at one time point, i.e. day 4. While this is understandable from the point of view of the numbers of hamsters that would need to be used to carry out comparison of multiple viruses at multiple time points, it has previously been reported that peak viral load is seen at 2 days post infection and that infectious viral load decreases by (Sia et al Nature 2020). This should be mentioned in the limitations of the study.”

We are thankful to Reviewer 2 for the comment and we agree that the inclusion of one time-point only is a limitation of the study and that this point should be elaborated more clearly in the manuscript. Inclusion of more

endpoints was indeed not possible due to the limitation of animal numbers and the original study design. Regarding the choice of the time-point, we agree with Reviewer 2 that from a virologic point of view, analysis of an earlier time-point would have been of advantage, since viral load tends to peak prior to 4dpi. However, the authors would like to point out that the main goal of the study was the assessment of histopathological lesions induced by VOC Omicron. The lesions induced by SARS-CoV-2 are in part manifestations and consequences of host responses, which lag viral replication and peak viral load. It has been previously reported that the peak of the extent and severity of the lesion in SARS-CoV-2 infected hamsters is observed between 3 and 6 dpi, depending on the study Imai et al., 2020 (DOI: 10.1073/pnas.2009799117); Osterrieder et al., 2020 (DOI: 10.3390/v12070779); Becker et al., 2021 (DOI: 10.3389/fimmu.2021.640842). Most importantly, the alveolar compartment usually does not show a prominent pathology before 3 dpi, since inflammation is mainly based on the conducting airways at early time-points. The assessment of alveolar damage is crucial for drawing conclusions on potential clinical significance. Based on the published data and the personal observations of the authors, 4 dpi represents a compromise that allows an optimal assessment of histopathology, in particular alveolar pathology, while still allowing the determination of the viral load. An additional paragraph has been added to the discussion to address the limitation and explain the rationale to the reader in more detail.

The following changes have been made:

DISCUSSION

-now lines 384-402 read as follows: “Since this study was performed with only one endpoint, it cannot be fully excluded that the observed milder lesions in VOC Omicron infected hamsters are the result of slower infection kinetics, rather than decreased pathogenicity. However, since recent studies reported that lung pathogenicity at 6 dpi¹⁷ or 7 dpi³¹ was significantly lower in VOC Omicron infected hamsters when compared to other VOCs, we believe that VOC Omicron-induced lung pathogenicity in our study is truly lower and not biased by the chosen time-point. The choice of 4 dpi as an endpoint for the comparison was based on the typical time course of SARS-CoV-2 infection in the hamster using ancestral strains. Previous studies have shown that viral load in the respiratory tract peaks at 2 dpi²³. However, the peak of lesion severity is observed between 3 and 6 dpi^{20,33,35,36}. As the primary aim of this study was to characterize histological lesions caused by VOC Omicron in comparison with those of other variants, the time-point chosen represents a compromise between assessment of histopathology and viral load. Although, one could argue that some of the observed mild lesions in the VOC Omicron group could be interpreted as background pathology, intralesional SARS-CoV-2 antigen found in most of these lesions confirms the causal role of VOC Omicron infection. Finally, it is important to note that we used 8-10 week old male hamsters. It was previously shown for the SARS-CoV-2 ancestral strain, that age- and sex-dependent differences in virus replication and shedding kinetics as well as lesion severity may exist^{15,33,47,48}. To what extent these factors may influence these parameters for infections with the respective SARS-CoV-2 variants is subject of future studies.”

4) “Line 65: typo. Should be “on” not “upon”

We sincerely apologize to Reviewer 2 for the shortcoming, now lines 65-66 read as follows: “The present study aims to obtain crucial information on the pathogenicity of VOC Omicron, focusing specifically on the respiratory tract”

5) “Line 152-153 appears redundant.”

We agree with Reviewer 2 comment former lines 152-153 have been removed from the main text.

6) “Lines 192/3. This sentence is confusing. What exactly are the authors trying to say? It may be relevant to say that the only statistically significant difference between other VOCs is seen in the % NP cells in the conductive airways where there are differences between Delta and Gamma, and 614G vs. Delta.”

We apologize to Reviewer 2 for the lack of clarity. The sentence has been rephrased accordingly, after correcting our statistical analysis for multiple comparisons, as suggested by Reviewer 3, using the Benjamini-Hochberg correction.

The following changes have been made:

-now lines 223-229 read as follows: “The amount of SARS-CoV-2 NP antigen in the pulmonary conductive airways of VOC Omicron infected hamsters was significantly lower compared to VOC Gamma and SARS-CoV-2 614G (Supplementary Table S4, Figure 7). In addition, SARS-CoV-2 614G infected animals showed significantly higher numbers of SARS-CoV-2 NP⁺ cells compared to VOC Delta (Supplementary Table S4). The amount of viral antigen in the lung parenchyma, which included the alveolar and vascular compartment, was significantly lower in VOC Omicron infected hamsters when compared to VOCs Gamma and Delta infected ones (Supplementary Table S4, Figure 7).”

7) “In figure S4, the conductive airways NP cells refers to what exactly? Is this the bronchiolar airways within the lung parenchyma as contrasted to the tracheal or bronchus?”

The authors thank Reviewer 2 for pointing out this shortcoming. In addition to the digital analysis description within the materials and methods section, where the definition of conductive airways is given, the authors now also included a brief footnote to Table S4 to improve clarity for the reader.

The following changes have been made:

SUPPLEMENTARY TABLE 4, footnote

Conductive airways: included bronchi, bronchioles and terminal bronchioles.

8) “Table S4. Significantly lower lung TCID50 with Omicron. This is comparable with the significantly lower viral titres seen with ex vivo cultures of the human lung lung with Omicron vs, Delta and wild type virus (Hui et al Nature 2022 doi: 10.1038/s41586-022-04479-6. PMID: 35104836).”

We thank Reviewer 2 for the valuable suggestion. The comment has been added to the discussion section of the manuscript.

The following changes have been made:

DISCUSSION

-now lines 346-353 read as follows (changes are underlined): “VOC Omicron infected animals also showed a lower number of immunolabeled cells in all pulmonary compartments and the lowest amount of viral RNA and infectious SARS-CoV-2 titers among all investigated groups. These lower lung viral titers are largely comparable to data obtained in an ex-vivo culture system of human lung tissue infected with multiple SARS-CoV-2 variants, including VOC Omicron³⁹. Moreover, our results are in agreement with a comparative study among SARS-CoV-2 variants, including VOC Omicron, in hamsters and mice¹⁷. Altogether, these results show a decreased pathogenicity of VOC Omicron for the lower respiratory tract of hamsters.”

Reviewer #3 (Remarks to the Author):

1) “The omicron variant of SARS-CoV-2 induces milder pathological changes in the upper and lower respiratory tract in Syrian golden hamsters than variants Gamma, Delta, or CoV-2 614G in male hamsters evaluated 4 days post-inoculation. The work will contribute to knowledge of the hamster model of SARS-CoV-2 infection to include in-depth reporting of histopathology associated with variants. It may help researchers by providing a baseline of expected changes at 4 DPI which could aid in developing treatments/preventative measures. Established literature has found similar results in this model but also often provides additional measures of virus measurements within tissues such as ISH or qRT-PCR for viral RNA. Additionally, published reports often provide information on infection at multiple time points post-inoculation and compare outcomes in males versus females.”

We thank Reviewer 3 for seeing in our work potential value for future scientific contributions in the field. As suggested, additional qRT-PCR analysis for viral load measurement were performed and included in the revised manuscript. The findings confirmed conclusions drawn from the results of virus titration and immunohistochemistry of nasal turbinates and lung. Moreover, study limitations pointed out by Reviewer 3, such as the lack of multiple time-points, are now mentioned in the discussion. We hope that the current version of the manuscript can be positively evaluated after adaptations based on valuable suggestions that will improve its scientific contents.

The following changes have been made:

DISCUSSION

-now lines 384-402 read as follows: “Since this study was performed with only one endpoint, it cannot be fully excluded that the observed milder lesions in VOC Omicron infected hamsters are the result of slower infection kinetics, rather than decreased pathogenicity. However, since recent studies reported that lung pathogenicity at 6 dpi¹⁷ or 7 dpi³¹ was significantly lower in VOC Omicron infected hamsters when compared to other VOCs, we believe that VOC Omicron-induced lung pathogenicity in our study is truly lower and not biased by the chosen time-point. The choice of 4 dpi as an endpoint for the comparison was based on the typical time course of SARS-CoV-2 infection in the hamster using ancestral strains. Previous studies have shown that viral load in the respiratory tract peaks at 2 dpi²³. However, the peak of lesion severity is observed between 3 and 6 dpi^{20,33,35,36}. As the primary aim of this study was to characterize histological lesions caused by VOC Omicron in comparison with those of other variants, the time-point chosen represents a compromise between assessment of histopathology and viral load. Although, one could argue that some of the observed mild lesions in the VOC Omicron group could be interpreted as background pathology, intralésional SARS-CoV-2 antigen found in most of these lesions confirms the causal role of VOC Omicron infection. Finally, it is important to note that we used 8-10 week old male hamsters. It was previously shown for the SARS-CoV-2 ancestral strain, that age- and sex-dependent differences in virus replication and shedding kinetics as well as lesion severity may exist^{15,33,47,48}. To what extent these factors may influence these parameters for infections with the respective SARS-CoV-2 variants is subject of future studies.”

2) “The work generally supports the conclusions however the lack of uninfected (ie mock-inoculated) controls limits ability to interpret the mild changes attributed to omicron. Procedural controls would enable the investigators to assess both baseline histology and any effect of intranasal inoculations on respiratory system independent of infection at a comparable time-point.”

The authors would like to thank Reviewer 3 for pointing out a limitation of the submitted study. The current study has been performed without a non-infected control group due to animal number restrictions and the original study design. As correctly pointed out by Reviewer 3, the lack of controls precludes an assessment of both baseline background pathological lesions and potential effects of intranasal inoculations unrelated to SARS-

CoV-2 infection. However, since viral antigen was detected within most of the lesions, we are confident that the potential overestimation of Omicron-induced lesions is minimal. Nevertheless, the point has been addressed in the revised version of the manuscript within the discussion where the main limitations of the study are mentioned and discussed.

The following changes have been made:

-now lines 384-402 read as follows: “Since this study was performed with only one endpoint, it cannot be fully excluded that the observed milder lesions in VOC Omicron infected hamsters are the result of slower infection kinetics, rather than decreased pathogenicity. However, since recent studies reported that lung pathogenicity at 6 dpi¹⁷ or 7 dpi³¹ was significantly lower in VOC Omicron infected hamsters when compared to other VOCs, we believe that VOC Omicron-induced lung pathogenicity in our study is truly lower and not biased by the chosen time-point. The choice of 4 dpi as an endpoint for the comparison was based on the typical time course of SARS-CoV-2 infection in the hamster using ancestral strains. Previous studies have shown that viral load in the respiratory tract peaks at 2 dpi²³. However, the peak of lesion severity is observed between 3 and 6 dpi^{20,33,35,36}. As the primary aim of this study was to characterize histological lesions caused by VOC Omicron in comparison with those of other variants, the time-point chosen represents a compromise between assessment of histopathology and viral load. Although, one could argue that some of the observed mild lesions in the VOC Omicron group could be interpreted as background pathology, intralesional SARS-CoV-2 antigen found in most of these lesions confirms the causal role of VOC Omicron infection. Finally, it is important to note that we used 8-10 week old male hamsters. It was previously shown for the SARS-CoV-2 ancestral strain, that age- and sex-dependent differences in virus replication and shedding kinetics as well as lesion severity may exist^{15,33,47,48}. To what extent these factors may influence these parameters for infections with the respective SARS-CoV-2 variants is subject of future studies.”

3) “SARS-CoV-2 Nucleoprotein was measured along with virus infectivity (TCID50) to compare levels of infectious virus in the nasal cavity and lungs. Assessing viral RNA via ISH or qRT-PCR could validate these conclusions.”

We thank Reviewer 3 for this valuable suggestion. As mentioned above (comment #1), qRT-PCR has been performed and included in the manuscript in order to validate our conclusions about viral load in the nasal turbinates and lung. The results were in line with virus titration and immunohistochemistry and confirmed that the viral load is lower in hamsters infected with Omicron compared to other strains and VOCs.

The following changes have been made:

RESULTS

-now lines 140-144 read as follows: “Quantification of SARS-CoV-2 RNA in the nasal turbinates confirmed the virus titration results. VOC Omicron infected hamsters showed the highest Ct values, corresponding to the lowest viral RNA copies. The Ct values were significantly higher compared to VOC Delta and Gamma. In addition, VOC Gamma infected hamsters showed significantly lower Ct values compared to 614G infected hamsters (Supplementary Table S2, Figure 3).”

-now lines 235-239 read as follows: “Quantification of SARS-CoV-2 RNA in the lung confirmed the virus titration results. VOC Omicron infected hamsters showed the highest Ct values, corresponding to the lowest viral RNA copies. The Ct values were significantly higher compared to VOC Gamma. In addition, VOC Gamma infected hamsters showed significantly lower Ct values compared to 614G and VOC Delta infected hamsters (Supplementary Table S4, Figure 7).”

MATERIALS AND METHODS

-now lines 468-475 read as follows:

“qRT-PCR

Viral RNA was extracted from nasal turbinates and lung using QIAmp Viral extraction kit according to the manufacturer’s instructions. RT-qPCR assay was performed using the protocol established by the Institut Pasteur⁵². In brief, primers and probe targeting SARS-CoV-2 E gene were used following the Super Script III Platinum One-Step RT-qPCR (Invitrogen) protocol. Amplification was performed as followed: reverse transcription 55 °C 20 min, denaturation 95 °C 3 min, followed by 50x cycles of amplification at 95 °C 15 s, 58 °C 30 s, where data was acquired. Further analysis and Cq values were determined using the Bio-Rad CFX Maestro software (BioRad). Analysis was performed in quadruplicates.”

4) *“The time-line of omicron disease progression may differ from the other variants, which could explain why it appears to be less severe at day 4 in this study that is focused on a single time point post-inoculation. In fact, the differences noted could be temporal rather than spatial. Additional time points should be evaluated to substantiate conclusions similar to recent reports in this model”*

We thank Reviewer 3 for pointing out this intriguing observation. Unfortunately, as mentioned before, limitations on animal numbers disabled us to use multiple time-points for comparison. As mentioned in comments #1 and #2, this limitation has been addressed in the revised version of the manuscript within the discussion where the main limitations of the study are mentioned and discussed. However, since recent studies reported that lung pathogenicity at 6 dpi Halfmann et al., 2022 (DOI: 10.1038/s41586-022-04441-6) or 7 dpi Suzuki et al., 2022 (DOI: 10.1038/s41586-022-04462-1) was significantly lower in VOC Omicron infected hamsters when compared to other VOCs, we are prone to believe that VOC Omicron-induced lung pathogenicity is truly lower and not biased by the chosen time-point. Nevertheless, trachea and nasal turbinates have not been analyzed at a later time-point in those studies.

The following changes have been made:

DISCUSSION

-now lines 384-402 read as follows: “Since this study was performed with only one endpoint, it cannot be fully excluded that the observed milder lesions in VOC Omicron infected hamsters are the result of slower infection kinetics, rather than decreased pathogenicity. However, since recent studies reported that lung pathogenicity at 6 dpi¹⁷ or 7 dpi³¹ was significantly lower in VOC Omicron infected hamsters when compared to other VOCs, we believe that VOC Omicron-induced lung pathogenicity in our study is truly lower and not biased by the chosen time-point. The choice of 4 dpi as an endpoint for the comparison was based on the typical time course of SARS-CoV-2 infection in the hamster using ancestral strains. Previous studies have shown that viral load in the respiratory tract peaks at 2 dpi²³. However, the peak of lesion severity is observed between 3 and 6 dpi^{20,33,35,36}. As the primary aim of this study was to characterize histological lesions caused by VOC Omicron in comparison with those of other variants, the time-point chosen represents a compromise between assessment of histopathology and viral load. Although, one could argue that some of the observed mild lesions in the VOC Omicron group could be interpreted as background pathology, intralesional SARS-CoV-2 antigen found in most of these lesions confirms the causal role of VOC Omicron infection. Finally, it is important to note that we used 8-10 week old male hamsters. It was previously shown for the SARS-CoV-2 ancestral strain, that age- and sex-dependent differences in virus replication and shedding kinetics as well as lesion severity may exist^{15,33,47,48}. To what extent these factors may influence these parameters for infections with the respective SARS-CoV-2 variants is subject of future studies.”

5) “Statistical analysis relies on pair-wise comparisons via Mann-Whitney tests without correcting appropriately for multiple group comparisons.”

We sincerely apologize to Reviewer 3 for the lack of statistical precision. Proper corrections for multiple group comparisons have been performed using the Benjamini-Hochberg correction. Therefore, some sentences pointing out to statistically significant results has been revised accordingly throughout the manuscript. Material and methods section, and all the graphs within the figures have been updated accordingly.

The following changes have been made:

MATERIAL AND METHODS

-now lines 531-537 read as follows:”

Statistical Analyses

Statistical analyses and graph design were performed using GraphPad Prism (GraphPad Software, San Diego, CA, USA) for Windows™. Data was tested for significant differences using Kruskal-Wallis tests. Pairwise comparisons among groups were obtained by two-tailed Mann-Whitney-U tests and corrected for multiple group comparisons using the Benjamini-Hochberg correction. Statistical significance was accepted at exact p-values of ≤ 0.05 (*), ≤ 0.01 (**), ≤ 0.001 (***) and ≤ 0.0001 (****), respectively.”

FIGURES 1-7

Graphs within Figure 1-7 have been changed accordingly and included in the revised version of the manuscript

6) “The interpretation that there is no viral spread to extra-respiratory organs based solely on IHC is questionable given sensitivity limitations of IHC. “

7) “Additionally, viral spread to other organs and development of lesions may occur at time points besides 4 days PI.”

We thank Reviewer 3 for pointing out these limitations. We agree that IHC is not the most sensitive method for virus detection and that low-level infection can be missed by the analysis. To support our conclusion by an additional method, virus titration of selected organs known as common extra respiratory localizations of SARS-CoV-2 detection (brain, kidney, intestine; Gupta et al., 2020 DOI: 202010.1038/s41591-020-0968-3; Munoz-Fontela et al., 2020 DOI: 10.1038/s41586-020-2787-6) has been performed in the mean time. No infectious virus was recovered from any of these organs. Nevertheless, the interpretation of the findings was rephrased, since the performed analysis cannot fully exclude that a transient or low-level spread can occur, possibly also at other time-points, in VOC Omicron infected hamsters.

The following changes have been made:

RESULTS

-Now lines 249-257 read as follows (changes are underlined): “The histopathological analysis of extra-respiratory organs sampled during necropsy of VOC Omicron infected hamsters such as brain, liver, spleen, kidney, adrenal gland, stomach, small and large intestine, pancreas and testicles revealed no significant lesions attributable to SARS-CoV-2. Importantly, SARS-CoV-2 NP antigen was not detected in any tissue outside the respiratory tract. Similarly, infectious virus was not isolated from selected organs which represent frequent localizations of extra-respiratory SARS-CoV-2 detection (brain, intestine, kidney)^{15,18}. Taken together, these results suggest that intranasal inoculation of hamsters with 10^4 TCID₅₀ of SARS-CoV-2 VOC Omicron variant does not cause infection and histopathological lesions in extra-respiratory organs at 4 dpi.”

DISCUSSION

-Now lines 375-383 read as follows: "SARS-CoV-2 infection of the CNS has been demonstrated in COVID-19 patients, murine models and in vitro in human brain organoids⁴⁴. Neuroinvasion through the olfactory route has also been demonstrated in hamsters, which was associated with neuroinflammation in the olfactory bulb²⁴. In the current study, neither viral antigen nor associated neuropathology or infectious virus were detected in the brain of VOC Omicron infected hamsters, probably due to the relatively limited infection observed in the olfactory mucosa. SARS-CoV-2 has also been reported to spread to extra-respiratory organs such as heart and kidney⁴⁵ or testes⁴⁶ in hamsters. Our hamsters did not show evidence of infection of extra-respiratory organs by histopathology, antigen detection and virus isolation, which may largely be due to the single time-point chosen for terminating the experiment."

8) "The semi-objective histologic scoring system is thorough and meets standards in the field however other groups have used more advanced and objective approaches to measure cellular immune responses in this model via digital image analysis."

We thank Reviewer 3 for considering our histopathology semi-quantitative scoring system adequate. In order to confirm our findings with a more objective and advanced approach, as suggested by Reviewer 3, we performed immunohistochemistry to detect and quantify macrophages/histiocytic cells (Iba-1) as well as neutrophils/heterophils (MPO), which are the most abundant cell types in the acute phase of infection. The immunolabeling was performed in all upper and lower respiratory tract organs. The quantification was performed by digital image analysis on whole slide scans of the respective organ sections. The numbers of immunopositive cells was quantified separately for distinct anatomical compartments. Overall, the results were largely in agreement with the findings obtained by semi-quantitative scoring of HE stained sections and showed that VOC Omicron infected animals have less inflammatory cell infiltrates in the upper and lower respiratory tract compared to other groups. In a few compartments, the quantitative analysis of the inflammatory cells population showed minor deviations from the results of semi-quantitative scoring. The authors believe that this minor deviation is not completely surprising taking into consideration that the immune cells are only one type change that has been evaluated with the HE semi-quantitative score system. For instance, the scoring also includes changes like hyperplasia, necrosis, edema, hemorrhage, intraluminal cell debris and fibrin exudation, that were also prominent in the nasal turbinates and in the lung. The extent of those changes does not always perfectly correlate with the degree of inflammatory cell numbers which could explain the minor variation observed. The authors believe that the new data complement the findings obtained by semi-quantitative scoring of lesions and enforce the overall conclusion of the manuscript. The new data and their interpretation have been included in the new version of the manuscript.

The following changes have been made:

RESULTS

-Now lines 99- 121 read as follows (changes are underlined): "

Rhinitis was moderate to severe marked in the majority of hamsters infected with either SARS-CoV-2 614G or VOCs Delta and Gamma. In contrast, hamsters infected with the VOC Omicron showed milder rhinitis. To confirm this, quantification of histopathological lesions was performed with a semi-quantitative scoring system, which includes the assessment of inflammation, hyperplasia, necrosis, and intraluminal cell debris. To confirm this, quantification of histopathological lesions was performed with a semi-quantitative scoring system. The overall nasal turbinate histopathological score of hamsters infected with VOC Omicron was significantly lower than that of hamsters infected with any other strain (Supplementary Table S2, Figure 2). Separate pathological evaluation of the respiratory and olfactory mucosa revealed that scores were significantly lower in both anatomical compartments in VOC Omicron infected hamsters compared to all other groups (Supplementary Table S2, Figure 2). Interestingly, SARS-CoV-2 614G infected animals showed significantly higher scores in the olfactory mucosa compared to VOCs Gamma and Delta (Supplementary Table S1). Histopathological semi-quantitative scores of

the nasal turbinates were mostly confirmed by the quantification of cells expressing myeloperoxidase (MPO, marker for neutrophils/heterophils) or ionized calcium-binding adapter molecule 1 (Iba-1, marker for macrophages and dendritic cells). In particular, VOCs Gamma and Delta infected hamsters showed significantly higher numbers of MPO+ cells in the total nasal mucosa compared to VOC Omicron infected hamsters. In the nasal respiratory mucosa, VOC Gamma infected hamsters revealed a significantly higher number of MPO+ cells compared to VOC Omicron infected hamsters. The nasal olfactory mucosa of VOC Omicron infected hamsters showed a lower number of MPO+ cells compared to VOCs Gamma and Delta as well as SARS-CoV-2 614G infected hamsters. Interestingly, the respiratory mucosa of VOC Omicron infected hamsters showed a significantly higher number of Iba-1+ cells compared to all the other groups, whereas the olfactory mucosa of VOC Omicron infected hamsters showed a significantly lower number of Iba-1+ cells (Supplementary table S2, Supplementary figure S1)."

FIGURES

- now Supplementary Figure 1 looks as follows:

Supplementary Figure S1. Decreased inflammatory cell infiltration in the nasal turbinates of VOC Omicron infected hamsters.

A) Representative images showing an overview of the caudal nasal turbinates of hamsters infected with VOCs Gamma, Delta, Omicron or 614G strain. The left panel shows hematoxylin and eosin stained sections. The middle and right panels show immunolabeling for myeloperoxidase (MPO, marker for neutrophils/heterophils) and ionized calcium-binding adapter molecule 1 (Iba1, marker for macrophages/histiocytic cells), respectively. The

inserts in the top panel show a detailed view of intraepithelial, immunopositive cells (brown signal). Omicron infected animals show lower numbers of immunoreactive cells in the caudal nasal turbinates, which are mostly covered by olfactory mucosa. Scale bars: 1 mm (overviews). B) Quantification of MPO+ in the total nasal mucosa as well as the respiratory and olfactory compartment shows lowest numbers of immunopositive cells in VOC Omicron infected hamsters. C) Quantification of Iba-1+ in the total nasal mucosa as well and in the olfactory compartment shows lowest numbers of macrophages/histiocytic cells in VOC Omicron infected hamsters. B, C) Data are shown as box and whisker plots with median and quartiles. Significant differences between the infection groups are indicated by * (* $p \leq 0.05$, ** $p \leq 0.01$). Data was tested by pairwise Mann-Whitney-U tests followed by Benjamini-Hochberg correction. N= 5 animals/group for VOCs Gamma and Delta and 6 animals/group for VOC Omicron and 614G. For quantification, one entire longitudinal section of the nasal turbinates was evaluated per animal. Cells were quantified exclusively within the mucosa, intraluminal debris was excluded from analysis.

-Now lines 154-172 read as follows (changes are underlined): “Tracheal lesions were observed in all SARS-CoV-2 infected hamsters and were characterized by multifocal to coalescing sub-epithelial infiltration with macrophages, lymphocytes and neutrophils/heterophils with frequent neutrophilic/heterophilic exocytosis. In addition, scattered single cell death and ciliary loss were observed. The severity of tracheitis varied from mild to moderate in individual animals, regardless of the SARS-CoV-2 variant used. Quantification of histopathological lesions was performed with a semi-quantitative scoring system, which includes an assessment of inflammation, hyperplasia, and intraluminal cell debris. The quantification of tracheal histopathology by semi-quantitative scoring showed no statistically significant differences among the groups (Supplementary Table S3, Figure 4). The results of histopathological scoring were confirmed by the quantification of immunolabeling for MPO+ and Iba-1+ cells. For both cell markers, no statistically significant differences were observed among the groups (Supplementary table S3, Supplementary figure S2). Based on these findings we further analyzed viral antigen expression in VOC Omicron infected animals comparing it with that of VOCs Gamma and Delta and SARS-CoV-2 614G. Viral antigen was exclusively detected in VOC Omicron infected hamsters, resulting in a statistically significant difference compared to all other groups. However, the number of positive cells was below 5% of all epithelial cells in all animals of this group. (Supplementary Table S3, Figure 5). Taken together, the trachea of VOC Omicron infected hamsters showed a mild to moderate tracheitis similar to the other groups, despite slightly higher numbers of SARS-CoV-2 NP+ cells.”

FIGURES

- now Supplementary Figure 2 looks as follows:

Supplementary Figure S2. Inflammatory cell infiltration in the trachea is comparable in hamsters infected with different strains of SARS-CoV-2.

A) Representative images showing tracheal mucosa of hamsters infected with VOCs Gamma, Delta, Omicron or 614G strain. The left panel shows hematoxylin and eosin stained sections. The middle and right panels show immunolabeling for myeloperoxidase (MPO, marker for neutrophils/heterophils) and Ionized calcium-binding adapter molecule 1 (Iba1, marker for macrophages/histiocytic cells), respectively. Hamsters infected with different SARS-CoV-2 strains show low numbers of MPO+ neutrophils/heterophils and moderate numbers of Iba-1+ macrophages/histiocytes in the epithelium and the lamina propria (brown signal). Scale bars: 50 μ m. B-C) Quantification of MPO (B) and Iba-1 (C) shows no significant differences among the infection groups. Data are shown as box and whisker plots with median and quartiles. Data was tested by pairwise Mann-Whitney-U tests followed by Benjamini-Hochberg correction. N = 5 animals/group for VOCs Gamma and Delta and 6 animals/group for VOC Omicron and 614G. For quantifications, one entire cross- and one entire longitudinal section were evaluated per animal.

-Now lines 189-217 read as follows (changes are underlined): "Pneumonia was moderate to marked in the majority of hamsters infected with either SARS-CoV-2 614G or VOCs Delta and Gamma. Quantification of histopathological lesions was performed with a semi-quantitative scoring system, which includes an assessment of inflammation, hyperplasia, necrosis, edema, hemorrhage and fibrin exudation. Interestingly, a milder pneumonia was consistently observed in hamsters infected with VOC Omicron. The overall lung histopathological score of VOC Omicron infected hamsters was significantly lower compared to VOCs Gamma and Delta and SARS-CoV-2 614G infected hamsters (Supplementary Table S4, Figure 6). Separate evaluation of the lesions in alveoli, conductive airways and vessels showed that the scores were significantly lower in all three compartments of VOC Omicron infected animals compared to the other groups (Supplementary Table S4, Figure

6). Interestingly, lesions in VOC Omicron infected hamsters, seemed to be mainly centered on the airways and the vascular compartment, while alveoli showed no or minimal involvement (Supplementary Figure 1). In contrast, hamsters infected with either SARS-CoV-2 614G or VOCs Gamma and Delta showed equal involvement of all compartments. Quantification of MPO+ and Iba-1+ cells largely confirmed histopathological scoring results. In particular, SARS-CoV-2 614G and VOC Omicron infected hamsters showed a significantly lower number of MPO+ cells in the whole lung compared to VOCs Gamma and Delta. In the alveolar compartment, SARS-CoV-2 614G and VOC Omicron infected hamsters revealed a significantly lower number of MPO+ cells compared to VOCs Delta and Gamma, or VOC Gamma infected hamsters, respectively. In the conductive airways, VOC Omicron infected hamsters had a significantly lower number of MPO+ cells compared to VOCs Gamma and Delta infected hamsters. In the vasculature compartment, SARS-CoV-2 614G showed a significantly lower number of MPO+ cells compared to VOCs Gamma and Delta infected hamsters. In addition, VOC Omicron infected hamsters showed a significantly lower number of Iba-1+ cells in the whole lung compared to VOCs Gamma and Delta. In the alveolar compartment, no statistically significant differences were observed among the groups, while in the conductive airways, VOC Omicron infected hamsters had a significantly lower number of Iba-1+ cells compared to VOC Gamma and SARS-CoV-2 614G. In the vascular compartment, VOC Omicron showed a significantly lower number of Iba-1+ cells compared to VOCs Gamma and Delta. (Supplementary table S4, Supplementary Figure S3)."

FIGURES

- now Supplementary Figure 3 looks as follows:

Supplementary Figure S3. VOC Omicron infected hamsters show decreased inflammatory cell infiltrates in the lung compared to hamsters infected with other SARS-CoV-2 strains.

A) Representative images showing the overview of left lung lobes of hamsters infected with VOCs Gamma, Delta, Omicron or 614G strain. The left panel shows hematoxylin and eosin stained sections. The middle and right panels show immunolabelings for myeloperoxidase (MPO, marker for neutrophils/heterophils) and Ionized calcium-binding adapter molecule 1 (Iba1, marker for macrophages/histiocytic cells), respectively. The inserts in the top panel show a detailed view of immunopositive cells (brown signal). B) Quantification of MPO in different lung compartments shows lower numbers of neutrophils/heterophils in hamsters infected with VOC Omicron or the 614G strain compared to hamsters infected with VOCs Gamma and Delta. C) Quantification of Iba-1 shows lowest numbers of macrophages/histiocytic cells in the total lung, conductive airways and vessels of VOC Omicron infected hamsters compared to other groups. Data are shown as box and whisker plots with median

and quartiles. Data was tested by pairwise Mann-Whitney-U tests followed by Benjamini-Hochberg correction. N = 5 animals/group for VOCs Gamma and Delta and 6 animals/group for VOC Omicron and 614G. For quantifications, one entire longitudinal section of the left lung lobe was evaluated.

MATERIAL AND METHODS

-Now lines 492-507 read as follows (changes are underlined):

“Immunohistochemistry

Immunohistochemistry of SARS-CoV-2 NP was performed using the Dako EnVision+ polymer system (Dako Agilent Pathology Solutions) and 3,3'-Diaminobenzidine tetrahydrochloride (Sigma-Aldrich, St. Louis, MO, United States) as previously described^{34,35}. Monoclonal mouse primary antibody against SARS-CoV-2 NP (Sino Biological, Peking, China-40143-MM05; dilution 1:16000) was applied overnight at 4°C. In order to immunolabel macrophages/histiocytes and heterophils/neutrophils, immunohistochemistry of Ionized calcium-binding adapter molecule 1 (Iba-1) and myeloperoxidase (MPO), respectively, was performed. The reaction was carried out using avidin–biotin complex (ABC) peroxidase kit (Vector Labs, Burlingame, CA, United States) and 3,3'-Diaminobenzidine tetrahydrochloride (Sigma-Aldrich, St. Louis, MO, United States) as previously described 53,35. Rabbit polyclonal primary antibodies against Iba-1 (FUJIFILM Wako Pure Chemical Corporation, Neuss, Germany; dilution 1:500) and MPO (Abcam, Cambridge, UK; dilution 1:200) was applied overnight at 4°C. For negative controls, specific primary antibodies were replaced by ascitic fluid from non-immunized BALB/cJ mice (for SARS-CoV-2 NP), and serum from non-immunized rabbits (for Iba-1 and MPO). The dilution of negative controls was chosen according to the protein concentration of replaced primary antibodies.”

MATERIAL AND METHODS

-Now lines 509-529 read as follows:

“Digital image analysis

For the quantification of immunopositive cells in nasal turbinates as well as in tracheal and pulmonary tissue, sections were digitized using the Olympus VS200 (Olympus Deutschland GmbH, Hamburg, Germany) slide scanner. Image analysis was performed using the open source software package QuPath for digital pathology image analysis⁵⁵. For all animals and all immunolabelings, nasal turbinates and tracheal whole slides images as well as one longitudinal section (along the main bronchus) of the entire left lung lobe were evaluated. In brief, regions of interest (ROI), in the nasal turbinates (respiratory and olfactory mucosa) and the trachea (tracheal epithelium and subepithelial layer) were identified by a pathologist. In particular for SARS-CoV-2 NP analysis in the lung, total lung tissue was detected automatically through digital thresholding and additional ROIs for conductive airways, including bronchi, bronchioles and terminal bronchioles were subsequently marked. Lung parenchyma (alveolar and vascular compartments) was then obtained by subtraction of conductive airways from total lung tissue. The total numbers of immunolabeled and non-labeled cells was determined by automated cell detection in all ROIs, based on the applied marker and tissue specific thresholding. For Iba-1 and MPO analysis in the lung, total lung tissue was detected automatically through digital thresholding and included additional ROIs for conductive airways, including bronchi, bronchioles and terminal bronchioles as well as vasculature t. Alveoli were obtained by subtraction of conductive airways and vasculature from total lung tissue. The total numbers of immunolabeled and non-labeled cells was determined by automated cell detection in all ROIs, based on marker and tissue specific thresholding.”

9) “Are there any images of the gross pathology? Do the lesions described correspond to histologic findings? Gross findings in these animals may be more related to peri-mortem changes consistent with euthanasia rather than viral infection.”

Unfortunately, gross pictures have not been taken of all animals, in part for logistic reasons. Retrospectively, this is a pity and would have improved data presentation. However, the authors would like to point out that the presentation of the histological lung lesions in the manuscript includes sub-gross pictures of the entire left lung lobe section (Figures 6 and 7), which gives the reader an overview of the differences in severity and distribution pattern of the lesions among the groups. The authors feel that the histopathological sub-gross pictures very specifically show virus-induced lesions similar to, and probably more detailed than gross pathology images. The gross pathology lesions described in the manuscript were largely corresponding with the histopathological findings. Occasionally, hyperemia and alveolar hemorrhages without inflammatory response were found on histology, but these lesions were not counted in the score because they were interpreted as peri-mortem changes and therefore not related to SARS-CoV-2 infection.

10) “Line 152-153: “After confirming the limited pathogenicity for the trachea, the investigations on the lower respiratory tract were completed with the lung investigations” - awkwardly worded and unclear”

We agree with Reviewer 3 comment, former lines 152-153 have been removed from the main text.

11) “Line 209 – which tissues were tested for NP antigen?”

The authors thank Reviewer 3 for pointing out this shortcoming. The authors listed also in the results section of the revised manuscript the tested extra-respiratory organs.

The following changes have been made:

RESULTS

-Now lines 249-257 read as follows (changes are underlined): “The histopathological analysis of extra-respiratory organs sampled during necropsy of VOC Omicron infected hamsters such as brain, liver, spleen, kidney, adrenal gland, stomach, small and large intestine, pancreas and testicles revealed no significant lesions attributable to SARS-CoV-2.”

12) “Line 380 – did all evaluators score all slides?”

The authors apologize to Reviewer 3 for the lack of clarity. The slides containing respiratory organs (nasal turbinates, trachea, lungs) were scored by two veterinary pathologists (FA and GB) and scoring was subsequently reviewed and confirmed by two board certified veterinary pathologists (MC and WB). Extra-respiratory organs were evaluated by one veterinary pathologist (LA) and confirmed by two board certified veterinary pathologists (MC and WB). The information has been added in the materials and methods section of the revised manuscript.

The following changes have been made:

MATERIALS AND METHODS

-now lines 476-491 read as follows:

“Histopathology

Formalin-fixed paraffin embedded samples were cut into 2 µm thick serial sections and stained with hematoxylin and eosin (H&E). Sections of the nasal turbinates, trachea, and lung were scanned using an Olympus VS200 Digital slide scanner (Olympus Deutschland GmbH, Hamburg, Germany) and evaluated in a blinded manner with a semi-quantitative scoring system with special emphasis on inflammation, degeneration and regeneration as previously described, with minor modification¹⁶. Histopathological semi-quantitative evaluations were performed by veterinary pathologists. In particular, nasal turbinates, trachea and lung slides were evaluated in a blinded fashion and scored by FA and GB. Extra-respiratory organs from VOC Omicron infected hamsters were evaluated by LA. Subsequently, histopathological evaluation and scoring were reviewed confirmed by board certified veterinary pathologists (WB, MC). Nasal turbinates were evaluated on a full length longitudinal section of the nose including respiratory and olfactory epithelium. Trachea was evaluated on cross- and longitudinal sections along the entire length of the organ. Finally, the lung was evaluated on one cross section (at the level of the entry of the main bronchus) and one longitudinal section (along the main bronchus) of the entire left lung lobe. The applied scoring systems are provided in details in Supplementary table 5.”

13) “Figure 2 Are all arrowheads meant to indicate intact olfactory epithelium? Confusing for images in Gamma and Delta. Also the labeling in the legend is not consistent with the image.”

The authors apologize for the lack of clarity. The images in the figure have been replaced (see response to Reviewer 1) and the labeling within the figure has been altered. Now, cell death is indicated by arrows and intraluminal debris is labelled with asterisks. The figure legend has been adapted to increase readability and consistency. In order to be consistent throughout the manuscript, the other figure legends have been changed accordingly.

The following changes have been made:

FIGURES

- now figure 2 looks as follows:

Figure 2: Decreased pathogenicity of VOC Omicron for nasal olfactory mucosa compared to other SARS-CoV-2 strains.

A) Representative images showing an overview of the caudal nasal turbinates (left panel) and high magnification of the olfactory mucosa (right panel) of hamsters infected with VOCs Gamma, Delta, Omicron or 614G strain. Olfactory mucosa in VOCs Gamma and Delta or 614G infected hamsters showed the most striking histopathological changes. Lesions were mainly characterized by disorganization, cell death (arrows) and cell sloughing, associated with intraluminal exudate composed of proteinaceous fluid, cell debris and degenerated neutrophils/heterophils (asterisks). Olfactory mucosa in VOC Omicron infected hamsters was largely intact and only occasional cell death was observed (arrow). Scale bars = 1 mm (overviews) and 50 μ m (high magnifications). Hematoxylin and eosin stain. B) Semi-quantitative analysis of nasal turbinates histopathology revealed significantly lower values in the overall score as well as the separate scores for respiratory and olfactory mucosa in VOC Omicron infected hamsters. Data are shown as box and whisker plots with median and quartiles. Significant differences between the infection groups are indicated by * ($p \leq 0.05$), ** ($p \leq 0.01$), **** ($p \leq 0.0001$). Data was tested by pairwise Mann-Whitney-U tests followed by Benjamini-Hochberg correction. N= 5 animals/group for VOCs Gamma and Delta and 6 animals/group for VOC Omicron and 614G. For quantification, one entire longitudinal section of the nasal turbinates was evaluated per animal.

14) "Figure 3 legend labeling not consistent with image"

The authors apologize for the lack of clarity. The images in the figure have been replaced (see response to Reviewer 1). The figure legend has been adapted to increase readability and consistency.

The following changes have been made:

FIGURES

- now Figure 3 looks as follows:

Figure 3: VOC Omicron infected hamsters showed decreased viral antigen and infectious viral titers in the nasal turbinates compared to other SARS-CoV-2 strains.

A) Representative images showing SARS-CoV-2 nucleoprotein (NP) immunolabeling in the nasal turbinates of hamsters infected with VOCs Gamma, Delta, Omicron or 614G strain. The images depict an overview of the caudal nasal turbinates (left panel) and high magnification of the olfactory mucosa (right panel). Viral antigen (brown signal) was mainly detected in epithelial cells within the olfactory mucosa and within intraluminal exudates (asterisks). Based on the cell morphology, characterized by apically located nuclei and abundant apical cytoplasm, most of these cells were sustentacular cells (arrowheads). The number of positive cells was highest in hamsters infected with 614 G, followed by hamsters infected with VOCs Gamma or Delta. In hamsters infected with VOC Omicron, only rare foci with a few positive cells were observed. Scale bars: 1 mm (overview) and 20

µm (high magnification). B) Quantification of SARS-CoV-2 NP antigen in the respiratory and olfactory mucosa. VOC Omicron infected hamsters displayed a significantly lower number of immunopositive cells in the olfactory mucosa compared to hamsters infected with 614G C) Quantification of infectious SARS-CoV-2 titers in the nasal turbinates. Titers were significantly lower in VOC Omicron infected hamsters compared to 614G and VOC Gamma infected hamsters. D) Quantification of SARS-CoV-2 RNA by qRT-PCR (Ct values). VOC Omicron infected hamsters show significantly higher Ct values, corresponding to low RNA content, compared to VOCs Gamma and Delta infected hamsters. B-D) Data are shown as box and whisker plots with median and quartiles. Significant differences between the infection groups are indicated by * (* p ≤ 0.05, ** p ≤ 0.01, *** p ≤ 0.001, **** p ≤ 0.0001). Data was tested by pairwise Mann-Whitney-U tests followed by Benjamini-Hochberg correction. N= 5 animals/group for VOCs Gamma and Delta and 6 animals/group for VOC Omicron and 614G. For quantifications, one entire longitudinal section of the nasal turbinates was evaluated per animal.

15) “Figure 6 legend line 615: “lungs of omicron infected animals were mildly affected with a marked affected and not in all lung compartments” – what is meant by this?”

The authors apologize for the lack of clarity. The figure legend has been revised.

The following changes have been made:

FIGURE LEGEND

- now lines 779-797 read as follows:

“Figure 6: Decreased pathogenicity of VOC Omicron in the lungs compared to other SARS-CoV-2 strains.

A) Representative images showing histopathological lesions in hamsters infected with VOCs Gamma, Delta, Omicron or 614G strain. The left panel shows an overview of the left lung lobe and the right panel higher magnifications of lesions in the conductive airways (CA), vessels (V) and surrounding alveoli. Lungs of hamsters infected with VOCs Gamma or Delta or the 614G strain showed the most extensive histopathological changes, which can be appreciated best on the overview pictures as consolidated, darker staining areas. Alveolar lesions were characterized by septal and luminal infiltration of macrophages and neutrophils/heterophils admixed with extravasated erythrocytes and fibrin, which obscured the alveolar architecture (arrows). Conductive airways frequently showed epithelial hyperplasia (arrowhead). Vascular lesions mainly consisted of histiocytic-neutrophilic/heterophilic perivascular and intramural infiltrates (asterisk o.ä.). Lungs of Omicron infected animals were only mildly affected in all lung compartments. Hematoxylin and eosin stain. Scale bars = 50 µm. **B)** Semi-quantitative analysis of pulmonary histopathology revealed significantly lower values in the overall lung score, lung conductive airway score, lung alveoli score, and lung vascular compartment score in VOC Omicron infected hamsters compared to the other investigated groups. Data are shown as box and whisker plots with median and quartiles. Significant differences between the infection groups obtained by Mann-Whitney-U tests are indicated by * (* p ≤ 0.05, ** p ≤ 0.01, *** p ≤ 0.001). N= 5 animals/group for VOCs Gamma and Delta and 6 animals/group for VOC Omicron and 614G. For quantifications, one entire cross and one entire longitudinal section of the left lung lobe were evaluated.”

16) “Scoring of trachea describes bronchi/bronchioles”

The authors apologize for the oversight. The respective section in supplementary table S5 has been revised.

The following changes have been made:

SUPPLEMENTARY MATERIAL

Table S5

1.1. Severity of tracheal inflammation (scored in area of maximal severity)		
0	No	No inflammation
1	Mild	Mild, mononuclear and granulocytic tracheitis (with exocytosis of inflammatory cells into epithelium, occasional single cell necrosis and mild, subepithelial infiltrates)
2	Moderate	Moderate, mononuclear and granulocytic to necrotizing tracheitis (with exocytosis of inflammatory cells into epithelium, frequent single cell necrosis and intraluminal debris and moderate, subepithelial infiltrates)
3	Severe	Marked, mononuclear and granulocytic and necrotizing tracheitis with widespread exocytosis of inflammatory cells into epithelium, frequent epithelial necrosis and intraluminal debris and severe, subepithelial infiltrates)

REVIEWERS' COMMENTS

Reviewer #1 (Remarks to the Author):

The authors have satisfactorily replied to most of the concerns.

Reviewer #2 (Remarks to the Author):

The revised manuscript has addressed the reviewer's concerns.

Reviewer #3 (Remarks to the Author):

The fundamental limitation of this manuscript remains evaluation of a single time point, day 4 pi. The authors themselves note that this timepoint is a compromise. They note that peak viral infection is likely earlier than day 4 while day 4 may be optimal for alveolar pathology - flanking timepoints are necessary in this study however to make these conclusions.

The study also still does not include procedural controls (uninfected) limiting ability to attribute mild lesions to infection versus intranasal instillation of mock-inoculum. Noting an association between histopathology alterations and any virus immunostaining is not indicative of causation.

Regarding discussion of kinetics of tracheal infection in the hamster model, Mulka et al 2022 (PMID: 34767812 PMID: PMC8577872 DOI: 10.1016/j.ajpath.2021.10.009) showed highest infectious virus titers in trachea at day 2 pi and then lower but detectable levels in trachea of most animals at day 4 pi. This contrasts with authors's statement that SARS-CoV-2 in trachea is transient and undetectable at day 3 in their experience.

Authors are responsive to other reviewer comments.

REVIEWER COMMENTS

Reviewer #1 (Remarks to the Author):

1) "The authors have satisfactorily replied to most of the concerns."

We thank Reviewer 1 for the positive evaluation and for the time dedicated to the review process that improved the quality of our work.

Reviewer #2 (Remarks to the Author):

1) "The revised manuscript has addressed the reviewer's concerns."

We thank Reviewer 2 for the positive evaluation and for the time dedicated to the review process that improved the quality of our work.

Reviewer #3 (Remarks to the Author):

- 1) ***“The fundamental limitation of this manuscript remains evaluation of a single time point, day 4 pi. The authors themselves note that this timepoint is a compromise. They note that peak viral infection is likely earlier than day 4 while day 4 may be optimal for alveolar pathology - flanking timepoints are necessary in this study however to make these conclusions.”***
- 2) ***“The study also still does not include procedural controls (uninfected) limiting ability to attribute mild lesions to infection versus intranasal instillation of mock-inoculum. Noting an association between histopathology alterations and any virus immunostaining is not indicative of causation.”***

We apologize to Reviewer 3 for the lack of clarity within the discussion of the manuscript. The part has been revised, pointing out the limitations of the current study more clearly.

The following changes have been made:

DISCUSSION

-now lines xxx read as follows: “The authors recognize that the study has some limitations. First, virological and histological studies have been carried out at one time-point only. This does not allow to draw final conclusions on the kinetics of VOC Omicron infection in comparison to other strains as well as to assess, whether the observed milder lesions in VOC Omicron infected hamsters are not the result of slower dynamics, rather than decreased pathogenicity. However, since recent studies reported that lung pathogenicity at 6 dpi¹⁷ or 7 dpi³¹ was significantly lower in VOC Omicron infected hamsters when compared to other VOCs, we believe that VOC Omicron-induced lung pathogenicity in our study is truly lower and not biased by the chosen time-point. The choice of 4 dpi as an endpoint for the comparison was based on the typical time course of SARS-CoV-2 infection in the hamster using ancestral strains. Previous studies have shown that viral load in the respiratory tract peaks at 2 dpi²³. However, the peak of lesion severity is observed between 3 and 6 dpi^{20,33,35,36}. As the primary aim of this study was to characterize histological lesions caused by VOC Omicron in comparison with those of other variants, the time-point chosen represents a compromise between assessment of histopathology and viral load. Future studies involving flanking time-points are warranted in order to corroborate the conclusions of the present study. Second, the experiment has been performed without a non-infected control group due to animal number limitations. One could argue that some of the observed mild lesions in the Omicron group could also be interpreted as background lesions or possible effects of intranasal inoculation. However, intralesional SARS-CoV-2 antigen was found in most foci with histopathological changes, suggesting an involvement of VOC Omicron.

Finally, it is important to note that we used 8-10 week old male hamsters. It was previously shown for the SARS-CoV-2 ancestral strain, that age- and sex-dependent differences in virus replication and shedding kinetics as well as lesion severity may exist^{15,33,47,48}. To what extent

these factors may influence these parameters for infections with the respective SARS-CoV-2 variants is subject of future studies.”

3) “Regarding discussion of kinetics of tracheal infection in the hamster model, Mulka et al 2022 (PMID: 34767812 PMID: PMC8577872 DOI:10.1016/j.ajpath.2021.10.009) showed highest infectious virus titers in trachea at day 2 pi and then lower but detectable levels in trachea of most animals at day 4 pi. This contrasts with authors's statement that SARS-CoV-2 in trachea is transient and undetectable at day 3 in their experience. “

We thank Reviewer 3 for the valuable comment. Additional recent studies (Mulka et al 2022, Schreiner et al 2022) have been added to the manuscript to complete the discussion about the kinetics of tracheal infection in the hamster model.

The following changes have been made:

DISCUSSION

-now lines xxx read as follows: “ Tracheal infection by non-Omicron strains has been reported by others at 4 dpi ^{24,30,31} by qRT-PCR or viral titration. In particular, a recent study showed highest infectious virus titers in trachea at 2 dpi and then lower levels in most animals at 4 dpi³¹. Another recent study, using SARS-CoV-2 NP immunolabeling found high numbers of SARS-CoV-2⁺ cells at 1 dpi, but only few immunolabelled cells at 3 dpi³². These findings are in agreement with previous studies that reported very low numbers of positive cells ^{29,33} or positive signal limited to detached cell debris³⁴ at 3-4 dpi with non-Omicron strains. Therefore, we assume that the trachea most likely was infected by non-Omicron strains, but that virus was largely or completely cleared below the detection level of immunolabeling at 4 dpi. ”

4) “Authors are responsive to other reviewer comments.”

We thank Reviewer 3 for the positive evaluation and for the time dedicated to the review process that improved the quality of our work.

REVIEWER COMMENTS

Reviewer #1 (Remarks to the Author):

1) "The authors have satisfactorily replied to most of the concerns."

We thank Reviewer 1 for the positive evaluation and for the time dedicated to the review process that improved the quality of our work.

Reviewer #2 (Remarks to the Author):

1) "The revised manuscript has addressed the reviewer's concerns."

We thank Reviewer 2 for the positive evaluation and for the time dedicated to the review process that improved the quality of our work.

Reviewer #3 (Remarks to the Author):

- 1) ***“The fundamental limitation of this manuscript remains evaluation of a single time point, day 4 pi. The authors themselves note that this timepoint is a compromise. They note that peak viral infection is likely earlier than day 4 while day 4 may be optimal for alveolar pathology - flanking timepoints are necessary in this study however to make these conclusions.”***
- 2) ***“The study also still does not include procedural controls (uninfected) limiting ability to attribute mild lesions to infection versus intranasal instillation of mock-inoculum. Noting an association between histopathology alterations and any virus immunostaining is not indicative of causation.”***

We apologize to Reviewer 3 for the lack of clarity within the discussion of the manuscript. The part has been revised, pointing out the limitations of the current study more clearly.

The following changes have been made:

DISCUSSION

-now lines xxx read as follows: “The authors recognize that the study has some limitations. First, virological and histological studies have been carried out at one time-point only. This does not allow to draw final conclusions on the kinetics of VOC Omicron infection in comparison to other strains as well as to assess, whether the observed milder lesions in VOC Omicron infected hamsters are not the result of slower dynamics, rather than decreased pathogenicity. However, since recent studies reported that lung pathogenicity at 6 dpi¹⁷ or 7 dpi³¹ was significantly lower in VOC Omicron infected hamsters when compared to other VOCs, we believe that VOC Omicron-induced lung pathogenicity in our study is truly lower and not biased by the chosen time-point. The choice of 4 dpi as an endpoint for the comparison was based on the typical time course of SARS-CoV-2 infection in the hamster using ancestral strains. Previous studies have shown that viral load in the respiratory tract peaks at 2 dpi²³. However, the peak of lesion severity is observed between 3 and 6 dpi^{20,33,35,36}. As the primary aim of this study was to characterize histological lesions caused by VOC Omicron in comparison with those of other variants, the time-point chosen represents a compromise between assessment of histopathology and viral load. Future studies involving flanking time-points are warranted in order to corroborate the conclusions of the present study. Second, the experiment has been performed without a non-infected control group due to animal number limitations. One could argue that some of the observed mild lesions in the Omicron group could also be interpreted as background lesions or possible effects of intranasal inoculation. However, intralesional SARS-CoV-2 antigen was found in most foci with histopathological changes, suggesting an involvement of VOC Omicron.

Finally, it is important to note that we used 8-10 week old male hamsters. It was previously shown for the SARS-CoV-2 ancestral strain, that age- and sex-dependent differences in virus replication and shedding kinetics as well as lesion severity may exist^{15,33,47,48}. To what extent

these factors may influence these parameters for infections with the respective SARS-CoV-2 variants is subject of future studies.”

3) “Regarding discussion of kinetics of tracheal infection in the hamster model, Mulka et al 2022 (PMID: 34767812 PMID: PMC8577872 DOI:10.1016/j.ajpath.2021.10.009) showed highest infectious virus titers in trachea at day 2 pi and then lower but detectable levels in trachea of most animals at day 4 pi. This contrasts with authors's statement that SARS-CoV-2 in trachea is transient and undetectable at day 3 in their experience. “

We thank Reviewer 3 for the valuable comment. Additional recent studies (Mulka et al 2022, Schreiner et al 2022) have been added to the manuscript to complete the discussion about the kinetics of tracheal infection in the hamster model.

The following changes have been made:

DISCUSSION

-now lines xxx read as follows: “ Tracheal infection by non-Omicron strains has been reported by others at 4 dpi ^{24,30,31} by qRT-PCR or viral titration. In particular, a recent study showed highest infectious virus titers in trachea at 2 dpi and then lower levels in most animals at 4 dpi³¹. Another recent study, using SARS-CoV-2 NP immunolabeling found high numbers of SARS-CoV-2⁺ cells at 1 dpi, but only few immunolabelled cells at 3 dpi³². These findings are in agreement with previous studies that reported very low numbers of positive cells ^{29,33} or positive signal limited to detached cell debris³⁴ at 3-4 dpi with non-Omicron strains. Therefore, we assume that the trachea most likely was infected by non-Omicron strains, but that virus was largely or completely cleared below the detection level of immunolabeling at 4 dpi. ”

4) “Authors are responsive to other reviewer comments.”

We thank Reviewer 3 for the positive evaluation and for the time dedicated to the review process that improved the quality of our work.